# Gaussian Ensemble Belief Propagation for Efficient Inference in High-Dimensional, Black-box Systems

**Dan MacKinlay**[*]
CSIRO's Data61
Dan.MacKinlay@data61.csiro.au

**Russell Tsuchida**
Monash University[†]

**Dan Pagendam**
CSIRO's Data61

**Petra Kuhnert**
CSIRO's Data61

## Abstract

Efficient inference in high-dimensional models is a central challenge in machine learning. We introduce the Gaussian Ensemble Belief Propagation (GEnBP) algorithm, which combines the strengths of the Ensemble Kalman Filter (EnKF) and Gaussian Belief Propagation (GaBP) to address this challenge. GEnBP updates ensembles of prior samples into posterior samples by passing low-rank local messages over the edges of a graphical model, enabling efficient handling of high-dimensional states, parameters, and complex, noisy, black-box generative processes. By utilizing local message passing within a graphical model structure, GEnBP effectively manages complex dependency structures and remains computationally efficient even when the ensemble size is much smaller than the inference dimension — a common scenario in spatiotemporal modeling, image processing, and physical model inversion. We demonstrate that GEnBP can be applied to various problem structures, including data assimilation, system identification, and hierarchical models, and show through experiments that it outperforms existing belief propagation methods in terms of accuracy and computational efficiency.

Supporting code is available at github.com/danmackinlay/GEnBP.

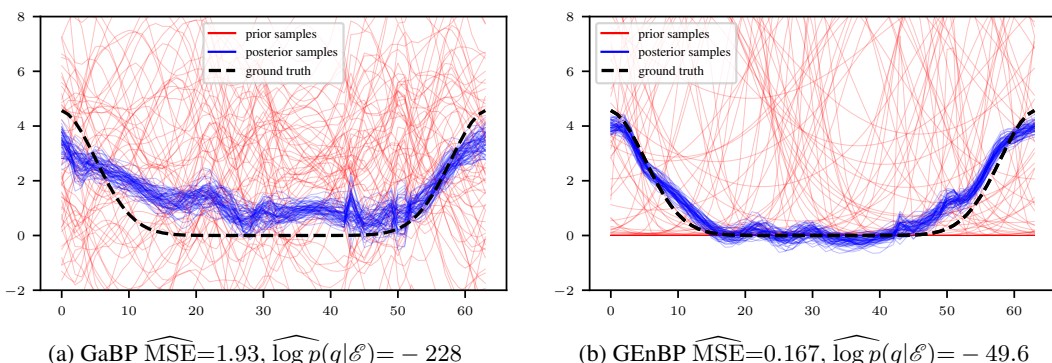

(a) GaBP $\widehat{\mathrm{MSE}}{=}1.93$, $\widehat{\log p(q|\mathscr{E})}{=}-228$

(b) GEnBP $\widehat{\mathrm{MSE}}{=}0.167$, $\widehat{\log p(q|\mathscr{E})}{=}-49.6$

Figure 1: Prior and posterior samples for latent **q** in the 1D system identification problem (Section 4.1). The GEnBP ensemble has size $N{=}64$, and $N{=}64$ samples are drawn from the GaBP posterior. The GEnBP prior comprises samples; the GaBP prior is sampled from a Gaussian density with the same moments.

---

[*]https://danmackinlay.name
[†]Work completed while at CSIRO's Data61

# 1 INTRODUCTION

We combine the Ensemble Kalman Filter (EnKF) and Gaussian Belief Propagation (GaBP) to perform inference in high-dimensional hierarchical systems. While both methods are well-established, their combination seems novel and empirically outperforms existing approaches in key problems.

GaBP (Yedidia et al., 2005) is a specific Gaussian message-passing algorithm designed for inference in graphical models. It performs density-based inference by leveraging a joint density over the model's state space. This density is approximated as a product of Gaussian factors, defining a graphical model (Koller & Friedman, 2009) over the graph $\mathcal{G}$. Sum-product messages passed along $\mathcal{G}$'s edges help infer target marginals (Pearl, 2008). Message-passing methods have been successfully applied in various applications, including Bayesian hierarchical models (Wand, 2017), error-correcting codes (Forney, 2001; Kschischang et al., 2001), and Simultaneous Localization and Mapping (SLAM) tasks (Dellaert & Kaess, 2017; Ortiz et al., 2021).

Graphical models possess various useful properties, such as permitting inference distributed over many computational nodes (Vehtari et al., 2019), domain adaptation (Bareinboim & Pearl, 2013), a natural means for estimating treatment effects (Pearl et al., 2016), and online, streaming updates (Dellaert & Kaess, 2017; Eustice et al., 2006). They perform well with low-dimensional variables but become computationally expensive in high-dimensional scenarios.

The EnKF (Evensen, 2003) is widely used for inference in state-space models. It constructs a posterior sample of a hidden state from prior samples by moment-matching the observation-conditional distribution. Although the state update relies on a Gaussian approximation, EnKF never explicitly evaluates the Gaussian density. This approach works well in high-dimensional settings, such as climate models (Houtekamer & Zhang, 2016), where computing densities can be prohibitively expensive. However, the standard EnKF is limited to state filtering and doesn't extend to more general inference tasks. In recent times, various techniques have extended EnKF to the system-identification setting, where not only states but latent parameters of dynamics are jointly inferred (e.g. Fearnhead & Künsch, 2018; Chen et al., 2023). EnKF is prized for its ability to handle correlated spatial fields efficiently, as with other spatial Gaussian approximations such as the Integrated Nested Laplace Approximation (Rue et al., 2009) and the related Stochastic Partial Differential Equation approach (Lindgren et al., 2011). Unlike those methods, it can exploit existing simulators to encode complex physical dynamics, and does not require the Jacobian of the simulator. Thus far, there are few methods which extend the attractive simulator-based, high-dimensional inference of the EnKF to the more general setting of graphical models.

GEnBP borrows strength from both EnKF and GaBP, achieving the EnKF's efficiency in high-dimensional data processing and GaBP's capability to handle complex graphical model structures (Table 1). GEnBP is capable of performing inference in models with noisy and moderately non-linear observation processes, unknown process parameters, and deeply nested dependencies among latent variables, scaling to millions of dimensions in variables and observations. The key insight is that, despite the name, Gaussian Belief Propagation as commonly used is not truly generic in the class of possible Gaussian approximations (Section 2.2.3). Like GaBP and EnKF, GEnBP uses Gaussian approximations. However, it relies on empirical samples for statistics over non-linear nodes, similar to EnKF, rather than the linearisation at the mode used in GaBP. By using ensembles to represent Gaussian distributions, GEnBP avoids the computational burden of full covariance matrices, enabling efficient belief propagation with potentially superior accuracy.

This approach is highly relevant for many practical problems, including physical and geospatial systems like computational fluid dynamics, geophysical model inversion, and weather prediction. Such systems typically feature high-dimensional representations with spatial correlations induced by known physical dynamics that create a low-rank structure, which empirical covariance approximations can effectively exploit.

**Main Contributions**

1. We introduce Gaussian Ensemble Belief Propagation (GEnBP), a novel message-passing method for inference in high-dimensional graphical models.

2. GEnBP leverages EnKF-like ensemble statistics instead of the traditional linearization used in GaBP, improving the handling of non-linear relationships and reducing computational complexity.

3. We demonstrate that GEnBP scales to higher dimensions than traditional GaBP while achieving comparable or better accuracy in practically important physical problems, such as spatiotemporal modeling and physical model inversion.

4. GEnBP is enabled by our development of:
   (a) rank-efficient techniques for propagating high-dimensional Gaussian beliefs efficiently, and
   (b) computationally efficient methods for converting between Gaussian beliefs and ensemble approximations without high-dimensional matrix operations,
   which are, to our knowledge, novel contributions in this context.

## 2  PRELIMINARIES

We introduce our notation and essential concepts. See Appendix C for belief propagation background and Appendix D for the Ensemble Kalman filter.

### 2.1  MODELS AS GENERATIVE PROCESSES AND DENSITIES

Random variables are denoted in sans-serif (e.g. $\mathsf{x}$) and their values in bold (e.g. $\boldsymbol{x}$); we assume all variables have densities, $\mathsf{x} \sim p(\boldsymbol{x})$.

The model state is partitioned into variables linked by a structural equation model $\mathcal{M}$ (Wright, 1934). Specifically, $\mathcal{M}$ comprises $J$ generating equations $\{\mathcal{P}_j\}_{j=1}^J$, where each $\mathcal{P}_j$ is defined as[1]

$$\mathcal{P}_j : \mathbb{R}^{D_{\mathscr{I}_j}} \to \mathbb{R}^{D_{\mathscr{O}_j}}, \quad \mathsf{x}_{\mathscr{I}_j} \mapsto \mathsf{x}_{\mathscr{O}_j}. \tag{1}$$

Each $\mathcal{P}_j$ relates an *input* set $\mathscr{I}_j$ to an *output* set $\mathscr{O}_j$. The *ancestral* variables, defined as $\mathscr{A} \coloneqq \bigcup_{\mathscr{I}_j=\emptyset} \mathscr{O}_j$, have no inputs. We classify variables as:

- *Evidence* $\mathscr{E}$: Observed variables.
- *Latent* $\mathscr{L}$: Unobserved or nuisance variables.
- *Query* $\mathscr{Q}$: Variables whose evidence-conditional distribution is of interest.

We assume that $\mathscr{Q} = \mathscr{A}$. This structure allows us to represent the joint density of all variables as a product of conditional densities,

$$p(\boldsymbol{x}) = \prod_{j=1}^J p(\boldsymbol{x}_{\mathscr{O}_j} \mid \boldsymbol{x}_{\mathscr{I}_j}). \tag{2}$$

Our main objective is specifically to compute the posterior distribution $p(\boldsymbol{x}_{\mathscr{Q}} \mid \mathsf{x}_{\mathscr{E}}=\boldsymbol{x}_{\mathscr{E}}^*) \propto \int p(\boldsymbol{x}_{\mathscr{Q}}, \boldsymbol{x}_{\mathscr{L}}, \boldsymbol{x}_{\mathscr{E}}=\boldsymbol{x}_{\mathscr{E}}^*)\mathrm{d}\boldsymbol{x}_{\mathscr{L}}.$. i.e. to update our beliefs about the ancestral variables by assimilating observations of the evidence variables while accounting for the latent variables.

### 2.2  BELIEF PROPAGATION IN GRAPHICAL MODELS

Graphical models associate a graph $\mathcal{G}$ with the factorization of the model density (see Equation 3). Here, BP refers to loopy belief propagation in factor graphs (Kschischang et al., 2001); additional details are in Appendix C.

The factor graph is constructed by representing each conditional probability $p(\boldsymbol{x}_{\mathscr{O}_j} \mid \boldsymbol{x}_{\mathscr{I}_j})$ as a *factor potential* $f_j$ over $\mathscr{N}_j \coloneqq \mathscr{O}_j \cup \mathscr{I}_j$:

$$p(\boldsymbol{x}) = \prod_j f_j\left(\boldsymbol{x}_{\mathscr{N}_j}\right). \tag{3}$$

$\mathcal{G}$ is bipartite, with factor nodes for each $f_j$ connected to eachh $\mathsf{x}_k$ in its neighborhood $\mathscr{N}_j$.

---

[1] Each $\mathcal{P}_j$ may be stochastic; we omit the noise terms for brevity.

BP approximates the marginal (or belief) of a query node as

$$b_{\mathcal{G}}\left(\boldsymbol{x}_k\right) \approx \int p(\boldsymbol{x})\,\mathrm{d}\boldsymbol{x}_{\setminus k}. \tag{4}$$

**Proposition 1** (Belief Propagation on Factor Graphs). *By iteratively propagating the messages,*

$$m_{f_j \to \mathbf{x}_k} = \int \left( f_j\left(\boldsymbol{x}_{\mathcal{N}_j}\right) \prod_{i \in \mathcal{N}_j \setminus k} m_{\mathbf{x}_i \to f_j} \right) \mathrm{d}\boldsymbol{x}_{\mathcal{N}_j \setminus k}, \tag{5}$$

$$m_{\mathbf{x}_k \to f_j} = \prod_{s \in \mathcal{N}_k \setminus j} m_{f_s \to \mathbf{x}_k}, \tag{6}$$

*BP approximates the marginals as*

$$b_{\mathcal{G}}\left(\boldsymbol{x}_k\right) = \prod_{s \in \mathcal{N}_k} m_{f_s \to \mathbf{x}_k} \approx \int p(\boldsymbol{x})\,\mathrm{d}\boldsymbol{x}_{\setminus k}. \tag{7}$$

*Proof:* See Yedidia et al. (2000). □

Though its analysis is complex (Yedidia et al., 2005; Weiss & Freeman, 2001), BP achieves state-of-the-art performance in many applications (Davison & Ortiz, 2019).

### 2.2.1 THE POSTERIOR GRAPH

In practice, we are interested in the evidence-conditional posterior graph $\mathcal{G}^*$, which conditions on $\mathbf{x}_{\mathcal{E}} = \boldsymbol{x}_{\mathcal{E}}^*$. To construct $\mathcal{G}^*$, we modify the factors $f_j$ for all $j \in \mathcal{N}_{\mathcal{E}}$ by replacing them with conditioned factors,

$$f_j(\boldsymbol{x}_j) \leftarrow f_j^*(\boldsymbol{x}_{\mathcal{N}_j \setminus \mathcal{E}}) \coloneqq p(\boldsymbol{x}_{\mathcal{N}_j \setminus \mathcal{E}} | \mathbf{x}_{\mathcal{E}} = \boldsymbol{x}_{\mathcal{E}}^*). \tag{8}$$

Observed variables and their edges are removed. The target marginal is then

$$p\left(\boldsymbol{x}_{\mathcal{Q}} \mid \mathbf{x}_{\mathcal{E}} = \boldsymbol{x}_{\mathcal{E}}^*\right) \approx b_{\mathcal{G}^*}\left(\boldsymbol{x}_{\mathcal{A}}\right),$$

with $b_{\mathcal{G}^*}(\boldsymbol{x}_{\mathcal{A}})$ the belief over ancestral variables. See Figure 11 in the Appendix.

### 2.2.2 NECESSARY OPERATIONS FOR BELIEF PROPAGATION

To compute $b_{\mathcal{G}^*}(\boldsymbol{x}_{\mathcal{A}})$, we perform conditioning, marginalization, and multiplication on densities:

**Definition 1.** *Consider a state space $\boldsymbol{x}^\top = \begin{bmatrix} \boldsymbol{x}_k^\top & \boldsymbol{x}_\ell^\top \end{bmatrix}$ with density $f(\boldsymbol{x}; \boldsymbol{\theta})$. The following operations suffice for BP in an observation-conditional graph ((Eq. 8)):*

$$\textbf{\textit{Conditioning:}} \quad f\left(\boldsymbol{x}; \boldsymbol{\theta}\right), \boldsymbol{x}_k^* \mapsto f^*\left(\boldsymbol{x}_\ell; \boldsymbol{\theta}_\ell^*\right) \coloneqq f\left(\boldsymbol{x}_\ell \mid \mathbf{x}_k = \boldsymbol{x}_k^*\right); \tag{9}$$

$$\textbf{\textit{Marginalization:}} \quad f\left(\boldsymbol{x}; \boldsymbol{\theta}\right) \mapsto f\left(\boldsymbol{x}_k; \boldsymbol{\theta}_k\right) \coloneqq \int f\left(\boldsymbol{x}; \boldsymbol{\theta}\right) \mathrm{d}\boldsymbol{x}_\ell; \tag{10}$$

$$\textbf{\textit{Multiplication:}} \quad f\left(\boldsymbol{x}; \boldsymbol{\theta}\right), f\left(\boldsymbol{x}_k; \boldsymbol{\theta}_k\right) \mapsto f\left(\boldsymbol{x}; \boldsymbol{\theta}'\right) \coloneqq f\left(\boldsymbol{x}; \boldsymbol{\theta}\right) f\left(\boldsymbol{x}_k; \boldsymbol{\theta}_k\right). \tag{11}$$

### 2.2.3 GAUSSIAN DISTRIBUTIONS IN BELIEF PROPAGATION

When variable relationships are linear with additive Gaussian noise, and ancestral distributions are Gaussian, each factor is Gaussian. In these cases, all operations in Definition 1 have closed-form solutions (Bickson, 2009; Yedidia et al., 2000), as used in Gaussian Belief Propagation (GaBP) (Dellaert & Kaess, 2017).

We represent Gaussian densities in two forms:

**Definition 2** (Gaussian Density Forms).

$$\textbf{\textit{Moments Form:}} \quad \phi_M(\boldsymbol{x}; \boldsymbol{m}, \mathrm{K}) = |2\pi\mathrm{K}|^{-\frac{1}{2}} \exp\left(-\frac{1}{2}(\boldsymbol{x} - \boldsymbol{m})^\top \mathrm{K}^{-1}(\boldsymbol{x} - \boldsymbol{m})\right), \tag{12}$$

$$\textbf{\textit{Canonical Form:}} \quad \phi_C(\boldsymbol{x}; \boldsymbol{n}, \mathrm{P}) = \left|\frac{\mathrm{P}}{2\pi}\right|^{\frac{1}{2}} \exp\left(-\frac{1}{2}\boldsymbol{x}^\top \mathrm{P}\boldsymbol{x} + \boldsymbol{n}^\top \boldsymbol{x} - \frac{1}{2}\boldsymbol{n}^\top \mathrm{P}^{-1}\boldsymbol{n}\right). \tag{13}$$

Here, $m$ and $\mathrm{K}$ are the mean and covariance, while $\mathrm{P} = \mathrm{K}^{-1}$ and $n = \mathrm{P}m$ are the precision and information vectors. Multiplication is simplest in canonical form:

$$\phi_C(\boldsymbol{x}; \boldsymbol{n}, \mathrm{P})\, \phi_C(\boldsymbol{x}_k; \boldsymbol{n}', \mathrm{P}') \propto \phi_C\left(\boldsymbol{x}; \boldsymbol{n} + \boldsymbol{n}', \mathrm{P} + \mathrm{P}'\right). \tag{14}$$

For nonlinear simulators $\mathcal{P}$, the joint covariance is non-Gaussian; standard GaBP methods (Eustice et al., 2006; Ranganathan et al., 2007; Dellaert & Kaess, 2017) then use the $\delta$-method (Dorfman, 1938) to approximate covariances via a first-order Taylor expansion:

$$\mathrm{Cov}(\mathbf{g}(\hat{\boldsymbol{\theta}})) \approx \mathbf{J}_g(\boldsymbol{\theta})\, \mathrm{Cov}(\hat{\boldsymbol{\theta}})\, \mathbf{J}_g(\boldsymbol{\theta})^\top,$$

where $\mathbf{J}_g$ is the Jacobian of $\mathbf{g}$ at $\boldsymbol{\theta}$.

The accuracy of the $\delta$-method is challenging to analyze for nonlinear $\mathcal{P}$. Additionally, GaBP scales unfavorably with the dimensionality of nodes, incurring a memory cost of $\mathcal{O}(D^2)$ and a time cost of $\mathcal{O}(D^3)$ whenever a $D \times D$ covariance matrix is inverted.

## 2.3 Ensemble Kalman Filtering

The Ensemble Kalman Filter (EnKF) reduces the computational burden of large $D$ by representing prior distributions with ensembles. An ensemble is a matrix of $N$ samples, $\mathrm{X} = [\boldsymbol{x}^{(1)}, \ldots, \boldsymbol{x}^{(N)}]$, with $\boldsymbol{x}^{(n)} \sim \phi_M(\boldsymbol{m}, \mathrm{K})$. We define the ensemble mean $\overline{\mathrm{X}} = \mathrm{XA}$ and deviation $\check{\mathrm{X}} = \mathrm{X} - \overline{\mathrm{X}}\mathrm{B}$, where $\mathrm{A} = \frac{1}{N}\mathbf{1}$ and $\mathrm{B} = \mathbf{1}^\top$.

By overloading the mean and variance operations to describe the empirical moments of ensembles, we define:

$$\widehat{\mathbb{E}}\mathrm{X} = \overline{\mathrm{X}}, \qquad \widehat{\mathrm{Var}}_V\mathrm{X} = \frac{1}{N-1}\check{\mathrm{X}}\check{\mathrm{X}}^\top + \mathrm{V}, \qquad \widehat{\mathrm{Cov}}(\mathrm{X}, \mathrm{Y}) = \frac{1}{N-1}\check{\mathrm{X}}\check{\mathrm{Y}}^\top \tag{15}$$

Here, $\mathrm{V}$ is a diagonal matrix known as the *nugget* term, typically set to $\sigma^2\mathrm{I}$. Setting $\sigma > 0$ is useful for numerical stability and to encode model uncertainty. These diagonal terms are usually treated as algorithm hyperparameters and also appear in GaBP.

The statistics of ensemble $\mathrm{X}$ define an implied Gaussian density:

$$\boldsymbol{x} \sim \phi_M(\boldsymbol{x}; \overline{\mathrm{X}}, \widehat{\mathrm{Var}}_V[\mathrm{X}]).$$

Prior model ensembles are sampled via ancestral sampling using the generative model described in Equation 25.

Two of the Belief Propagation (BP) operations from Definition 1 are applicable to the EnKF.

**Proposition 2.** *Partition* $\mathbf{x}^\top = \begin{bmatrix} \mathbf{x}_k^\top & \mathbf{x}_\ell^\top \end{bmatrix}$ *such that* $\mathrm{X}^\top = \begin{bmatrix} \mathrm{X}_k^\top & \mathrm{X}_\ell^\top \end{bmatrix}$. *Assume the ensemble* $\mathrm{X}$ *follows the Gaussian distribution:*

$$\mathrm{X} \sim \phi_M\left(\begin{bmatrix} \boldsymbol{x}_k \\ \boldsymbol{x}_\ell \end{bmatrix}; \begin{bmatrix} \overline{\mathrm{X}}_k \\ \overline{\mathrm{X}}_\ell \end{bmatrix}, \begin{bmatrix} \widehat{\mathrm{Var}}_V\mathrm{X}_k & \widehat{\mathrm{Cov}}(\mathrm{X}_\ell, \mathrm{X}_k) \\ \widehat{\mathrm{Cov}}(\mathrm{X}_k, \mathrm{X}_\ell) & \widehat{\mathrm{Var}}_V(\mathrm{X}_\ell) \end{bmatrix}\right). \tag{16}$$

*In ensemble form, conditioning (Eq. 9) is performed as:*

$$\mathrm{X}, \boldsymbol{x}_k^* \mapsto \mathrm{X}_\ell + \widehat{\mathrm{Cov}}(\mathrm{X}_\ell, \mathrm{X}_k)\widehat{\mathrm{Var}}_V^{-1}(\mathrm{X}_k)(\boldsymbol{x}_k^*\mathrm{B} - \mathrm{X}_k) \tag{17}$$

*The computational cost of solving Equation 17 is* $\mathcal{O}(N^3 + N^2 D_{\mathbf{x}_k})$. *Marginalization (Eq. 10) is simply truncation, i.e.,* $\mathrm{X} \mapsto \mathrm{X}_k$.

*Proof:* See Appendix D. $\qquad\qquad \square$

While the EnKF reduces computational costs and potentially improves approximation accuracy over GaBP for nonlinear relationships, it does not generalise to other model structures without the multiplication operation (Eq. 11) needed for BP.

In the following sections, we extend the EnKF-like approach to the BP setting by defining the necessary operations to handle general graphical models.

Table 1: Relations in Gaussian Ensemble Belief Propagation

| | **Generative** | **Density-based** |
|---|---|---|
| Operations | • Sample 
 • Condition | • Propagate |
| Graph type | Directed | Factor |
| Decomposition | $\mathbf{x}_3 = \mathcal{P}(\mathbf{x}_1, \mathbf{x}_2)$ | $f(x_1, x_2, x_3)$ |
| Node Parameters | Empirical moments 
 $\boldsymbol{m}, \mathrm{K}$ | Canonical parameters 
 $\boldsymbol{n}, \mathrm{P}$ |

## 3 GAUSSIAN ENSEMBLE BELIEF PROPAGATION

GEnBP consists of: (1) sampling from the generative prior; (2) converting ensemble statistics to canonical form; (3) performing BP with low-rank representations; and (4) recovering ensemble samples to match updated beliefs (see Figure 1). Although conceptually simple, the implementation details are delicate (see Appendix H).

We consider an approximation $\phi_M(\hat{\boldsymbol{m}}, \hat{\mathrm{K}})$ optimal if $\hat{\boldsymbol{m}} = \boldsymbol{m}$ and the Frobenius norm $\|\mathrm{K} \hat{-} \mathrm{K}\|_F$ is minimized—equivalently, if the Maximum Mean Discrepancy (MMD) with a second-order polynomial kernel is minimized (Sriperumbudur et al., 2010, Example 3).

### 3.1 RANK-EFFICIENT GAUSSIAN PARAMETERISATIONS

A key component of GEnBP is the use of *Diagonal Matrix with Low-rank perturbation* (DLR) representations. We represent a symmetric positive-definite matrix $\mathrm{K} \in \mathbb{R}^{D \times D}$ in DLR form as

$$\mathrm{K} = \mathrm{V} + s\,\mathrm{LL}^\top,$$

with $\mathrm{L} \in \mathbb{R}^{D \times N}$, diagonal V, and $s \in \{-1, +1\}$ (see Appendix F). In the EnKF, the empirical covariance from ensemble samples is naturally DLR; here, $N$ denotes the ensemble size (or rank of the DLR factors) and $D$ the variable dimension.

To initialize BP, we set each factor $f_j^*$ to the empirical distribution of the ensemble at that factor, $\mathrm{X}_j$:

$$f_j^* \sim \phi_M(\boldsymbol{x}; \overline{\mathrm{X}}_j, \widehat{\mathrm{Var}}_{\gamma^2 \mathrm{I}} \mathrm{X}_j). \tag{18}$$

Among possible DLR approximations, the ensemble-based approach is most favorable; see Appendix I for a cost comparison.

**Proposition 3.** *Suppose* $\mathbf{x} \sim \phi_M(\boldsymbol{x}; \boldsymbol{m}, \mathrm{K})$ *has a DLR covariance* $\mathrm{K} = \mathrm{LL}^\top + \mathrm{V}$, *with* $\mathrm{L} \in \mathbb{R}^{D \times N}$ *and* V *diagonal. Then, the canonical form parameters* $\boldsymbol{n}$ *and* P *can be computed efficiently:*

$$\boldsymbol{n} = \mathrm{P}\boldsymbol{m}, \quad \mathrm{P} = \mathrm{K}^{-1} = \mathrm{U} - \mathrm{RR}^\top,$$

*where* U *is diagonal and* $\mathrm{R} \in \mathbb{R}^{D \times N}$. *Both the conversion to canonical form and the recovery of moments have time complexity* $\mathcal{O}(N^3 + N^2 D)$.

*Proof:* Standard application of the Woodbury identity. See Appendix F.3. $\qquad\square$

## 3.2 Belief Propagation with DLR Representations

Standard GaBP requires expensive full-matrix operations for density multiplication. In contrast, GEnBP leverages the DLR structure for efficiency. For two DLR matrices,

$$\mathrm{K} = \mathrm{V} + s\,\mathrm{LL}^\top, \quad \mathrm{K}' = \mathrm{V}' + s\,\mathrm{L}'\mathrm{L}'^\top,$$

their sum is

$$\mathrm{K} + \mathrm{K}' = (\mathrm{V} + \mathrm{V}') + s\,[\mathrm{L} \quad \mathrm{L}']\,[\mathrm{L} \quad \mathrm{L}']^\top.$$

Thus, density multiplication (cf. Equation 14) can be performed efficiently while maintaining the DLR representation. Other operations such as vector products, rank reduction, and marginalisation are also efficient (see Appendix F).

## 3.3 Ensemble Conformation

After message propagation, we update ensemble samples to reflect the new beliefs (ensemble conformation) by finding an affine transformation of the prior ensemble that best matches the posterior mean and covariance. Specifically, we seek an affine transformation

$$T_{\boldsymbol{\mu},\mathrm{T}} : \mathrm{X} \mapsto \boldsymbol{\mu}\mathrm{B} + \check{\mathrm{X}}\mathrm{T},$$

with $\boldsymbol{\mu} = \boldsymbol{m}$ (the posterior mean) and $\mathrm{T} \in \mathbb{R}^{N \times N}$ chosen to minimize

$$\mathrm{T} := \underset{\mathrm{T}^*}{\arg\min}\left\|\widehat{\mathrm{Var}}_{\eta^2\mathrm{I}}(\boldsymbol{\mu}\mathrm{B} + \check{\mathrm{X}}\mathrm{T}^*) - (\mathrm{LL}^\top + \mathrm{V})\right\|_F^2. \tag{19}$$

Since $\mathrm{T}$ is identifiable only up to a unitary transform, we have:

**Proposition 4.** *Any symmetric positive semi-definite matrix* $\mathrm{G} = \mathrm{TT}^\top$ *satisfying*

$$\check{\mathrm{X}}^\top\check{\mathrm{X}}\,\mathrm{G}\,\check{\mathrm{X}}^\top\check{\mathrm{X}} = (N-1)\left(\check{\mathrm{X}}^\top\mathrm{L}(\check{\mathrm{X}}^\top\mathrm{L})^\top - \check{\mathrm{X}}^\top(\mathrm{V} - \eta^2\mathrm{I})\check{\mathrm{X}}\right)$$

*defines an affine transformation that minimizes the loss in Equation 19. Such a* $\mathrm{G}$ *can be computed with memory complexity* $\mathcal{O}(M^2 + N^2)$ *and time complexity* $\mathcal{O}(N^3 + DN^2 + DM^2)$.

*Proof:* See Appendix H.3. $\qquad\qquad\square$

For a variable with $K$ neighbors, in the worst case $M = KN$ and the ensemble recovery cost is $\mathcal{O}(K^3N^3 + DN^2K^2)$.

## 3.4 Computational Complexity

The overall cost of GEnBP depends on the graph structure and ensemble size $N$. In many applications $N \ll D$, yielding significant savings compared to GaBP, which scales poorly with $D$. Table 2 summarizes the computational complexities of GaBP and GEnBP. Notably, GEnBP avoids the $\mathcal{O}(D^3)$ costs of full covariance operations, making it well-suited for high-dimensional problems.

GEnBP scales as $\mathcal{O}(D)$ with $D$ (versus $\mathcal{O}(D^3)$ for GaBP) but as $\mathcal{O}(K^3)$ with node degree (versus $\mathcal{O}(1)$). In practice, large factors can be decomposed via Forney factorization (Forney, 2001; de Vries & Friston, 2017) to reduce $K$ without altering marginal distributions. If the $K$ scaling is prohibitive, one may convert to a full covariance matrix and use GaBP updates (see Appendix I); however, in our experiments this was unnecessary.

## 4 Experiments

In this section, we compare GEnBP against an alternative belief propagation method, GaBP, and, for reference, a global Laplace approximation (Mackay, 1992). We use synthetic benchmarks designed to assess performance in high-dimensional, nonlinear dynamical systems. In both cases, the graph structure is a randomized system identification task (Appendix A.1), where a static parameter influences a noisily observed, nonlinear dynamical system. Although GEnBP applies beyond system identification, this benchmark is chosen for its simplicity and popularity. More sophisticated graphical

Table 2: Computational complexities for Gaussian Belief Propagation (GaBP) and Gaussian Ensemble Belief Propagation (GEnBP), where $D$ is the node dimension, $K$ the node degree, and $N$ the ensemble size. Time complexities are measured in floating-point multiplications.

| Operation | GaBP | GEnBP |
|---|---|---|
| **Time Complexity** | | |
| Simulation | $\mathcal{O}(1)$ | $\mathcal{O}(N)$ |
| Error propagation | $\mathcal{O}(D^3)$ | — |
| Jacobian calculation | $\mathcal{O}(D)$ | — |
| Covariance matrix | $\mathcal{O}(D^2)$ | $\mathcal{O}(ND)$ |
| Factor-to-node message | $\mathcal{O}(D^3)$ | $\mathcal{O}(K^3N^3 + DN^2K^2)$ |
| Node-to-factor message | $\mathcal{O}(D^2)$ | $\mathcal{O}(1)$ |
| Ensemble recovery | — | $\mathcal{O}(K^3N^3 + DN^2K^2)$ |
| Canonical-Moments conversion | $\mathcal{O}(D^3)$ | $\mathcal{O}(N^3 + N^2D)$ |
| **Space Complexity** | | |
| Covariance matrix | $\mathcal{O}(D^2)$ | $\mathcal{O}(ND)$ |
| Precision matrix | $\mathcal{O}(D^2)$ | $\mathcal{O}(NDK)$ |

model problems are discussed in Appendix C, and Appendix B.3 demonstrates GEnBP applied to few-shot domain adaptation of a system dynamics emulator on unobserved states.

For $t = 1, 2, \ldots, T$ we define

$$\mathbf{x}_t = \mathcal{P}_{\mathbf{x}}(\mathbf{x}_{t-1}, \mathbf{q}), \qquad\qquad \mathbf{y}_t = \mathcal{P}_{\mathbf{y}}(\mathbf{x}_t) \qquad (20)$$

where the evidence is $\mathscr{E} = \{\mathbf{y}_t = \mathbf{y}_t^*\}_{t=1}^T$ and the query is $\mathscr{Q} = \mathbf{q}$.[2]

The GaBP implementation is modified from Ortiz et al. (2021) to allow arbitrary prior covariance. Hyperparameters are not directly comparable between methods; we address this by choosing favorable values for each algorithm. We measure performance by the mean squared error (MSE) of the posterior mean estimate relative to the ground truth $q_0$ and by the log-likelihood $\log b_{\mathcal{G}^*}(q_0; \mathbf{q})$ at $q_0$. In gradient-based methods such as Laplace or GaBP, the estimated posterior covariance may not be positive definite, rendering the log-likelihood undefined; in such cases, it is omitted. Additional experiments and details are provided in Appendix B.

## 4.1  1D TRANSPORT MODEL

The transport problem (Appendix A.2) is a simple nonlinear dynamical system defined over a 1-dimensional spatial extent, chosen for ease of visualization. States are subject to both transport and diffusion, with the transport term introducing nonlinearity. Observations consist of subsampled state vectors perturbed by additive Gaussian noise. Figure 1 shows sample prior and posterior distributions produced by GaBP and GEnBP, demonstrating that GEnBP recovers the posterior substantially better. Further analysis of this example is provided in Appendix A.2.1, where it is also compared against a global Laplace approximation.

## 4.2  2D FLUID DYNAMICS MODEL

In the computational fluid dynamics (CFD) problem (Appendix A.3), states are governed by discretized Navier–Stokes equations over a 2D spatial domain. The parameter of interest is a static, latent forcing field $\mathbf{q}$. In this setting, GaBP fails for $D_{\mathscr{Q}} > 1024$, and the Laplace approximation is unavailable because the estimated posterior covariance is far from positive definite. In our 2D incompressible model, both the state and forcing fields are scalar fields over a $d \times d$ domain, which we stack into vectors so that $D_{\mathscr{Q}} = D_{\mathbf{x}_t} = d^2$. For comparison with a classic sampler that does not use belief propagation, we implement a Langevin Monte Carlo (LMC) algorithm (Roberts & Tweedie, 1996) as a baseline, running it until it achieves an MSE comparable to that of GEnBP (2000 iterations after a 1000-iteration burn-in).

Figure 2 shows results as the dimension increases. The Langevin Monte Carlo algorithm achieves the best posterior RMSE, but at roughly $10^3$ times the computational cost of GEnBP. GEnBP attains

---

[2]If $\mathbf{q}$ were known, estimating $\{\mathbf{x}_t\}_t$ from $\{\mathbf{y}_t\}_t$ would be a filtering problem, solvable by EnKF.

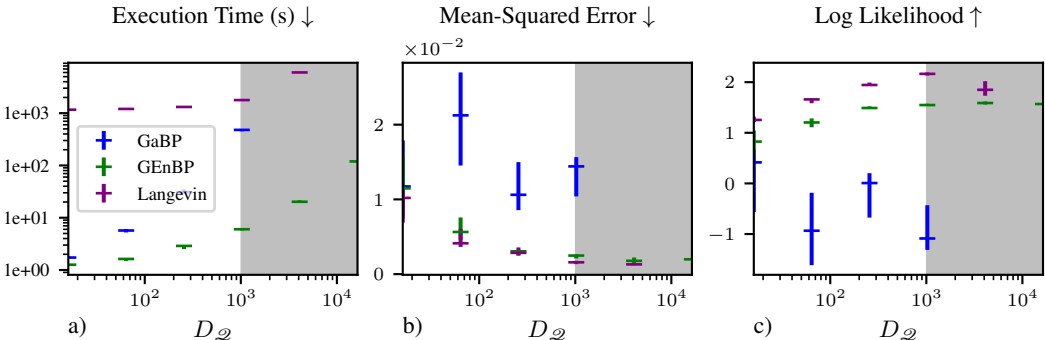

Figure 2: Influence of dimension $D_{\mathscr{Q}}$ in the 2D fluid dynamics model. Error bars indicate empirical 50% intervals from $n = 10$ runs. In grey-shaded regions, GaBP ran out of memory.

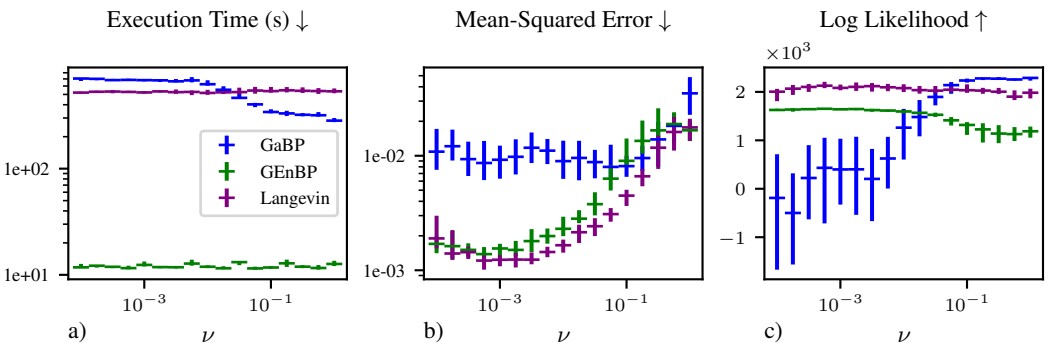

Figure 3: Influence of viscosity $\nu$ on a $32 \times 32$ 2D fluid dynamics model. Error bars indicate empirical 50% intervals from $n = 40$ simulations.

superior MSE and posterior likelihood compared to GaBP while scaling to a much higher $D_{\mathscr{Q}}$. GaBP experiments are truncated because trials with $D_{\mathscr{Q}} > 1024$ failed due to resource exhaustion.

We caution that absolute runtimes should be interpreted carefully. In particular, the high-dimensional Jacobian calculation in GaBP is not effectively parallelized by the PyTorch library, penalizing GaBP by a constant factor. The asymptotic $\mathcal{O}(D)$ scaling of these Jacobian calculations (Margossian, 2019) will eventually favor GEnBP.

Figure 3 illustrates the effect of varying viscosity $\nu$ from laminar to turbulent regimes, producing diverse nonlinear behaviors (see Figure 6). GEnBP strictly dominates GaBP in speed. It generally outperforms in terms of MSE, although there are ranges around $\eta \approx 1$ where performance is similar or slightly worse. Regarding the posterior log-likelihood at the ground truth, GEnBP is superior in the low-$\nu$ (turbulent) regime, while GaBP performs better in the high-$\nu$ (laminar) regime. Langevin Monte Carlo is not competitive in terms of posterior log-likelihood, although its performance might be improved through hyperparameter tuning.

## 5 CONCLUSIONS

Gaussian Ensemble Belief Propagation (GEnBP) advances inference in probabilistic graphical models by combining the strengths of the Ensemble Kalman Filter (EnKF) and Gaussian Belief Propagation (GaBP). It scales to higher factor dimensions than GaBP and achieves superior accuracy in complex, high-dimensional problems. By employing ensemble approximations, GEnBP accommodates the larger and more intricate factors common in real-world applications. Unlike EnKF, GEnBP handles complex dependencies without requiring gradients and often outperforms GaBP in both accuracy and speed.

While GEnBP introduces additional tuning parameters and requires more model executions during sampling, these costs are often offset by avoiding the high-dimensional Jacobian calculations needed in GaBP. Its effectiveness is best in systems where the dynamics can be well-approximated in a low-dimensional subspace (i.e., via DLR approximations). As with GaBP and EnKF, GEnBP relies on Gaussian approximations and is thus constrained to unimodal distributions; convergence analysis remains an open question. Moreover, comparing overall computational cost is complex and depends on the graph structure. Our empirical results demonstrate that there are problems where GEnBP is significantly more efficient than GaBP, although the full range of such problems has not been characterized.

Despite these limitations, GEnBP's scalability, flexibility, and ease of use make it a promising tool for applications such as geospatial prediction and high-dimensional data assimilation. Future work will focus on integrating practical improvements from both GaBP and EnKF—such as covariance localization, covariance inflation, adaptive ensemble selection, message damping, incremental updating, robust covariance scaling, and graph truncation—to enhance stability and convergence at lower cost. Exploring connections with particle-based and score-based methods may further expand GEnBP's capabilities.

## 6  ACKNOWLEDGEMENTS

We are indebted to Laurence Davies, Rui Tong, Cheng Soon Ong and Hadi Afshar for helpful discussions and feedback on this work.

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

## A    BENCHMARK PROBLEMS

To demonstrate our method's utility, we consider example problems of the system identification type, where a latent parameter must be inferred from its influence on the dynamics of a hidden Markov model. In this context, the problem is closely related to—but more challenging than—state filtering.

Unless otherwise specified, the hyperparameters for both GaBP and GEnBP are set to $\gamma^2 = 0.01$ and $\sigma^2 = 0.001$. GEnBP has additional hyperparameters $\eta^2 = 0.1$ and ensemble size $N = 64$. We cap the number of message-propagation descent iterations at 150 and relinearise or re-simulate after every 10 steps.

### A.1    SYSTEM IDENTIFICATION

In the system identification problem, our goal is to estimate the time-invariant latent system parameter $\mathbf{q}$. This parameter influences all states in the hidden Markov model, where $\mathbf{x}_0$ represents the unobserved initial state and subsequent states $\mathbf{x}_1, \mathbf{x}_2, \ldots$ evolve over time. Each state $\mathbf{x}_i$ is associated with an observation $\mathbf{y}_i$, as depicted in Figure 4.

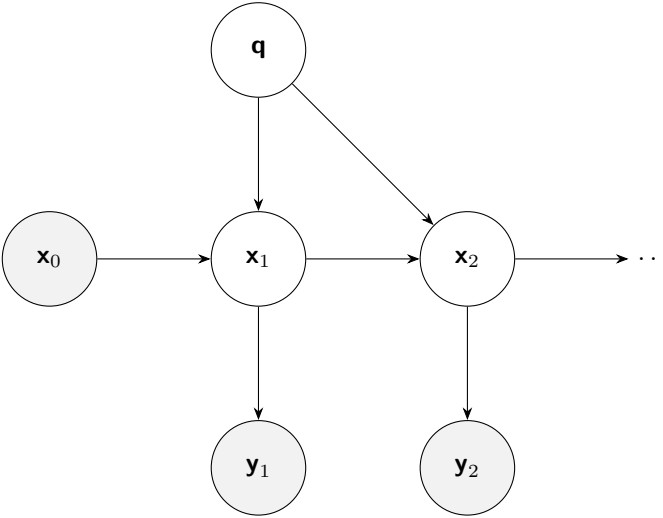

Figure 4: Generative model for the system identification problem. The latent parameter $\mathbf{q}$ influences all states, and observed states are shaded.

### A.2    ONE-DIMENSIONAL TRANSPORT PROBLEM

We consider a one-dimensional state-space model where the state at time $t$, $\mathbf{x}_t \in \mathbb{R}^d$, evolves according to

$$\mathcal{P}_t^x : \mathbf{x}_t \mapsto \mathbf{x}_{t+1}$$
$$= S_k\Big(C * \big(\gamma\mathbf{x}_t + (1-\gamma)\mathbf{q}\big)\Big) + \boldsymbol{\epsilon}_t, \tag{21}$$

$$\mathcal{P}_t^y : \mathbf{x}_t \mapsto \mathbf{y}_t$$
$$= D_\ell\big(\mathbf{x}_t\big) + \boldsymbol{\eta}_t, \tag{22}$$

where:

- $\mathbf{x}_t$ is the state vector at time $t$.
- $\mathbf{q} \in \mathbb{R}^d$ is a fixed parameter vector to be estimated.
- $\gamma \in [0, 1]$ is a decay parameter.
- $C * \mathbf{z}$ denotes convolution of $\mathbf{z}$ with a kernel $C$ (modeling diffusion).
- $S_k$ is a circular shift operator shifting the vector by $k$ positions (modeling advection).
- $\boldsymbol{\epsilon}_t \sim \mathcal{N}(0, \tau^2 I_d)$ is process noise.

- $\mathbf{y}_t$ is the observation vector at time $t$.
- $D_\ell$ is a downsampling operator selecting every $\ell$-th element.
- $\boldsymbol{\eta}_t \sim \mathcal{N}(0, \sigma^2 I_m)$ is observation noise.

The latent parameter $\mathbf{q}$ is drawn from a prior that generates smooth periodic functions. Specifically, each element is defined as

$$q_k = A \exp\Big(\kappa \cos\Big(\frac{2\pi k}{d} - \mu\Big)\Big), \quad k = 0, 1, \ldots, d-1, \tag{23}$$

where

- $\mu \sim \text{Uniform}[0, 2\pi]$ is a random phase.
- $\kappa \sim \chi^2\Big(\text{df} = \text{smoothness} \cdot d^{-1/2}\Big)$ controls the concentration (i.e. smoothness) of the function.
- $A$ is a scaling constant.

**Operator Definitions**

- **Convolution** ($C * \mathbf{z}$): Applies a blur kernel $C$ to the vector $\mathbf{z}$, modeling diffusion.
- **Circular Shift** ($S_k$): Shifts $\mathbf{z}$ by $k$ positions to the right, wrapping around.
- **Downsampling** ($D_\ell$): Selects every $\ell$-th element from $\mathbf{z}$, reducing spatial resolution.

**Intuitive Interpretation**  The state update in Equation 21 models a transport process where the state $\mathbf{x}_t$ evolves by:

1. Decaying toward a background field: $\gamma \mathbf{x}_t + (1 - \gamma)\mathbf{q}$.

2. Diffusing via convolution with $C$.

3. Being advected via the circular shift $S_k$.

4. Incorporating process noise $\boldsymbol{\epsilon}_t$.

The observation equation Equation 22 represents measurements of the state at reduced resolution with additive noise $\boldsymbol{\eta}_t$. Our inference goal is to estimate $\mathbf{q}$ from the sequence of observations $\{\mathbf{y}_t\}$ and the initial state $\mathbf{x}_0$. We use $T = 10$ timesteps for this problem.

### A.2.1 RESULTS

Figure 5 plots the influence of the state dimension $D_{\mathscr{Q}}$ on various inference quality measures. GEnBP estimates are substantially more accurate than GaBP, as indicated by lower MSE and more stable posterior likelihoods. In this relatively low-dimensional problem, a global Laplace approximation is also available. We observe that while the Laplace approximation is faster than both GaBP and GEnBP, its MSE performance is intermediate, and its posterior likelihood is less stable—especially as $D_{\mathscr{Q}}$ increases beyond 100. In contrast, GEnBP's posterior log likelihood degrades more slowly and remains defined.

### A.3 NAVIER–STOKES SYSTEM

The Navier–Stokes equation is a classic problem in fluid modeling (Ferziger et al., 2019). Here we consider 2D incompressible flow solved with a spectral method using a PyTorch implementation from Li et al. (2020). Defining the vorticity $\omega = \frac{\partial v}{\partial x} - \frac{\partial u}{\partial y}$ and streamfunction $\psi$ via

$$u = \frac{\partial \psi}{\partial y},$$

$$v = -\frac{\partial \psi}{\partial x},$$

the Navier–Stokes equations are:

1. Poisson equation: $\nabla^2 \psi = -\omega$.

2. Vorticity equation: $\frac{\partial \omega}{\partial t} + u\frac{\partial \omega}{\partial x} + v\frac{\partial \omega}{\partial y} = \nu \nabla^2 \omega$.

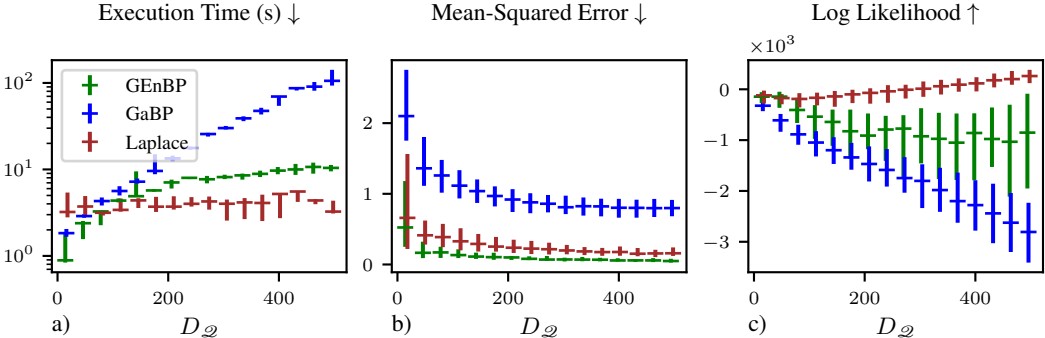

Figure 5: Influence of dimension $D_{\mathcal{Q}}$ in the transport example. Error bars are empirical 90% intervals from $n = 40$ runs. Missing log likelihood values indicate undefined values from non-positive definite covariance estimates.

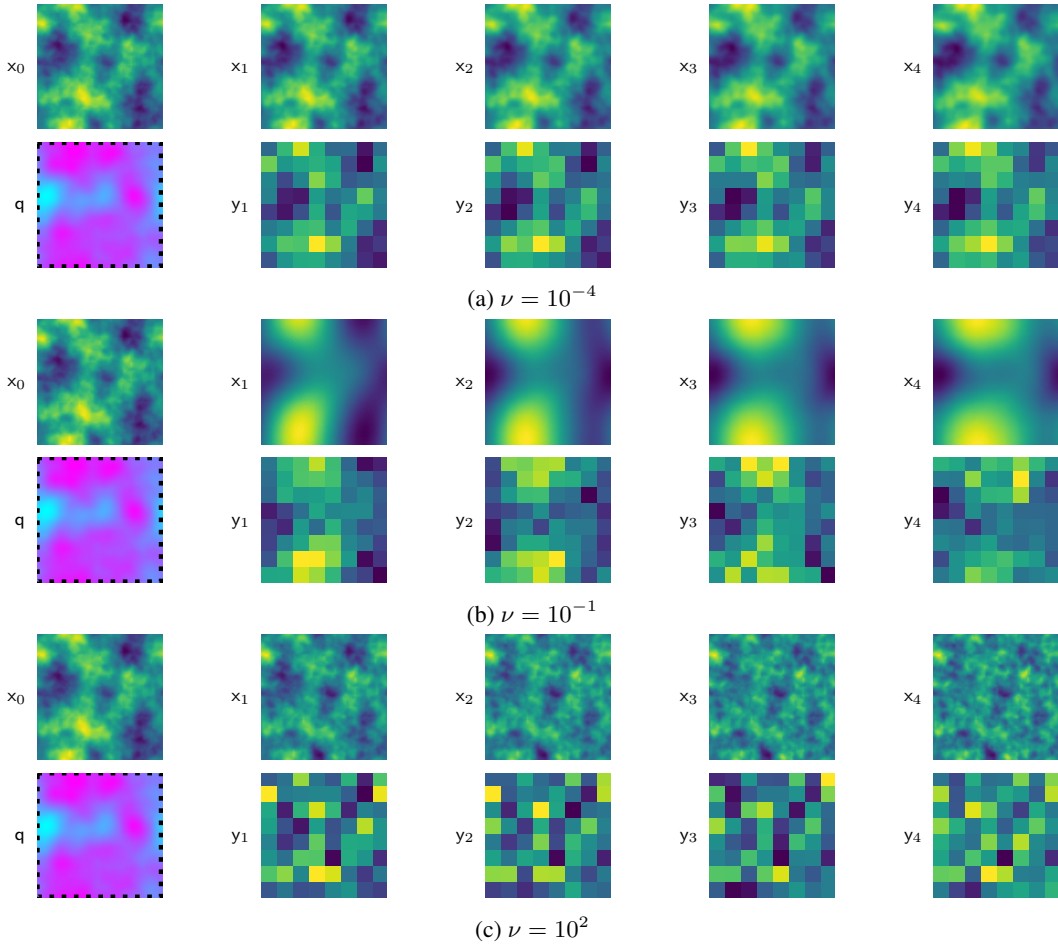

Figure 6: Three Navier–Stokes simulations with $\Delta t = 0.2$ and varying viscosity $\nu$. The simulation is on a $64 \times 64$ grid; observations are $8 \times 8$ and corrupted by white Gaussian noise with standard deviation $\sigma = 0.1$ on a normalized scale.

The equations are discretized onto a $d \times d$ grid. At each timestep, we inject additive Gaussian white noise $\boldsymbol{\nu}_t \sim \mathcal{N}(\mathbf{0}, \sigma_{\boldsymbol{\eta}}^2 I_{d^2})$ and a static forcing term $\mathbf{q}$ to the velocity field. The state $\mathbf{x}_t$ is obtained by stacking the 2D streamfunction $\psi$ into a vector. Observations are given by

$$\mathbf{y}_t = \mathrm{downsample}_\ell(\mathbf{x}_t) + \boldsymbol{\eta},$$

with $\boldsymbol{\eta} \sim \mathcal{N}(\mathbf{0}, \sigma_{\boldsymbol{\eta}}^2 I)$.

The prior state and forcing are sampled from a discrete periodic Gaussian random field with power spectral density

$$PSD(k) = \sigma^2 \big(4\pi^2 \|k\|^2 + \tau^2\big)^{-\alpha}. \tag{24}$$

## B EXPERIMENTAL DETAILS AND ADDITIONAL RESULTS

### B.1 HARDWARE CONFIGURATION

Experiments were conducted on a Dell PowerEdge C6525 Server with AMD EPYC 7543 32-Core Processors running at $2.8\,\mathrm{GHz}$ ($3.7\,\mathrm{GHz}$ turbo) with $256\,\mathrm{MB}$ cache. Float precision is set to 64 bits, and memory usage is capped at $32\,\mathrm{GB}$. Execution time is limited to 119 minutes.

### B.2 ENSEMBLE SIZE

In the main text, we leave the choice of ensemble size $N$ open. In practice, ensembles on the order of $N \approx 10^2$ seem sufficient. The computational cost of GEnBP scales as $\mathcal{O}(DN^2 + N^3)$ (see Section 3.4), but the extra precision gained by increasing $N$ is not unbounded. Figure 7 shows an experiment where $N$ is varied from 16 to 512 in steps of 16. The performance improves rapidly up to $N = 64$ with diminishing returns thereafter. We use $N = 64$ as our default.

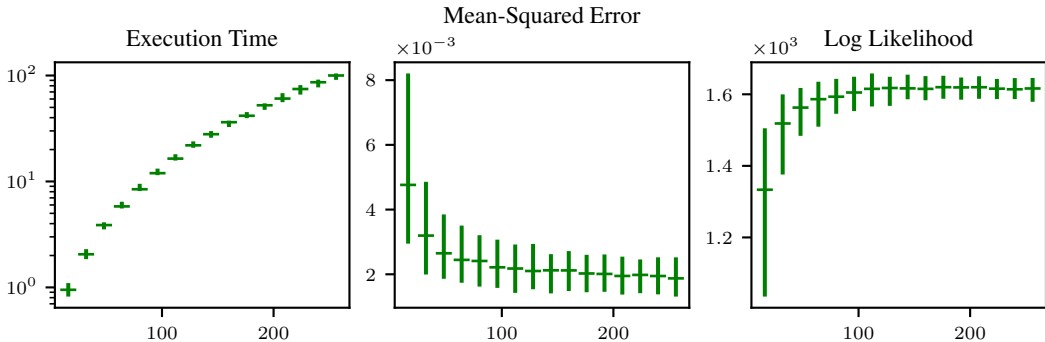

Figure 7: Performance of GEnBP versus ensemble size $N$ for the 1D transport system identification problem (Appendix A.2). Confidence intervals are empirical 95% intervals based on $n = 80$ runs.

### B.3 EMULATION USING GEnBP

In Section 4, we demonstrated GEnBP in system identification problems, which are pedagogically useful but may not showcase the full range of belief propagation methods. GEnBP can, however, be applied to more complex problems.

Here, we devise a proof-of-concept experiment in few-shot domain adaptation of a neural network emulator for unobserved states. The graphical model structure is shown in Figure 10.

The interpretation is as follows: We have a forward model $\mathcal{P} : \mathbf{x}_{t-1}, \mathbf{u} \mapsto \mathbf{x}_t$, a known observation operator $\mathcal{H} : \mathbf{x}_t \mapsto \mathbf{y}_t$, and a neural network $\mathcal{Q} : \mathbf{x} | \mathbf{u}$ trained to predict the state given the unknown parameter $\mathbf{u}$. In the system identification problem, we stop here. However, in few-shot domain adaptation, we introduce a Bayesian neural network $\mathcal{N} : \hat{\mathbf{x}}_{t-1}, \mathbf{w} \mapsto \mathbf{x}_t$ with weights $\mathbf{w}$ previously trained on related (possibly synthetic) data. Our goal is to adapt the neural network to a new target problem by estimating the posterior distribution $p(\mathbf{w}|\mathbf{y})$ using GEnBP.

In our experiment, we use the CFD model from Section 4 as the ground truth. For the neural network, we employ the Fourier Neural Operator (FNO) architecture (Li et al., 2020) because it can produce high-fidelity results with fewer than $10^6$ parameters. We simulate 10 time steps on a $32 \times 32$ grid with viscosity $\nu = 0.001$ and $\Delta t = 0.1$. GEnBP is applied to the first 5 time steps for domain adaptation, with observations downsampled by a factor of 5 and corrupted with zero-mean Gaussian noise ($\sigma_{\text{obs}}^2 = 0.01$). An ensemble size of $N = 30$ is used. The FNO employs 12 Fourier modes, 32 hidden channels, and 4 layers, with a Tensor Fourier Neural Operator variant that incorporates a 5% rank factor, totaling 1,590,030 weights. Figure 8 shows that the domain-adapted emulator closely tracks the target density over subsequent timesteps, while the unadapted emulator diverges.

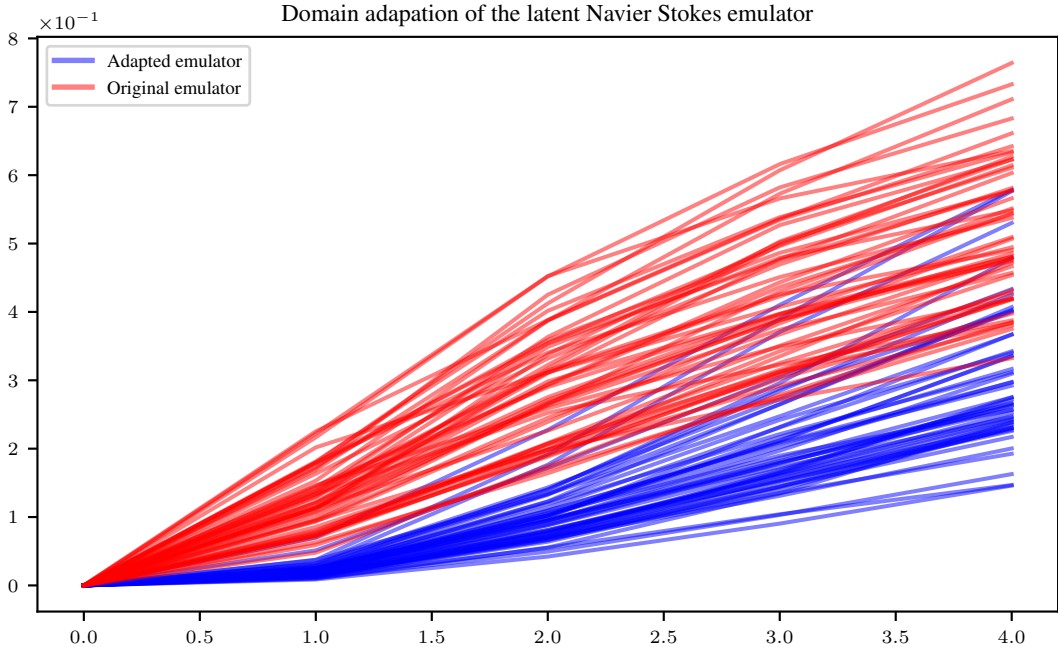

Figure 8: Performance of GEnBP in the emulator domain adaptation problem. The graph shows the Maximum Mean Discrepancy (MMD) between the observation-conditional state distribution, the unadapted emulator, and the domain-adapted emulator, computed using an isotropic Gaussian kernel of width $1/\sqrt{D}$, over 50 randomized emulation problems.

## C   MINIMUM VIABLE INTRODUCTION TO PROBABILISTIC GRAPHICAL MODELS

The field of probabilistic graphical models is mature and extensive (see, e.g., Koller & Friedman (2009); Wainwright & Jordan (2008)). For our purposes, it suffices to understand that this framework enables inference in complex hierarchical systems with partial observations. For example, Figure 9 illustrates a simplified weather model where an oceanic physics simulator and an atmospheric physics simulator provide forward predictions and interact through physical coupling. Their outputs are observed by different sensors (thermal IR satellite imagery for the oceans and phased radar for the atmosphere). These systems are high-dimensional, noisy, and governed by nonlinear PDEs. Our primary target here is the one-step-ahead prediction for the atmospheric model, though extensions to jointly learn observation and physics parameters are straightforward.

Figure 10 shows a model closer to the system identification problem described in Section 4. In this model, we extend the system identification task (recovering $\mathbf{u}$) by also recovering an emulator (e.g., a neural network) that predicts an approximate state $\hat{\mathbf{x}}_t$ from the previous emulated state $\hat{\mathbf{x}}_{t-1}$. The goal is to estimate a posterior over neural network weights $\mathbf{w}$ to improve predictions when only noisy observations $\mathbf{y}_t$ are available.

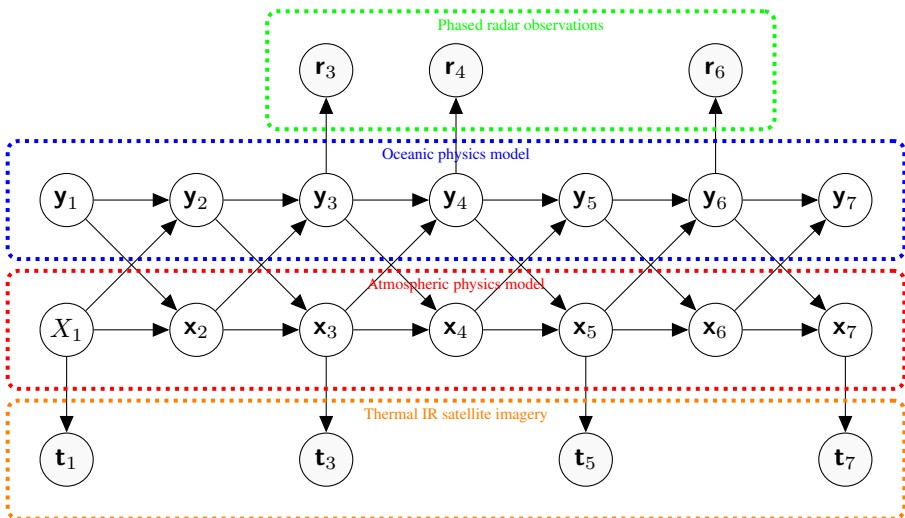

Figure 9: A complex graphical model structure from a weather prediction problem. It includes an oceanic physics simulator, an atmospheric physics simulator, phased radar observations, and thermal IR satellite imagery. The model is high-dimensional, noisy, and governed by nonlinear partial differential equations.

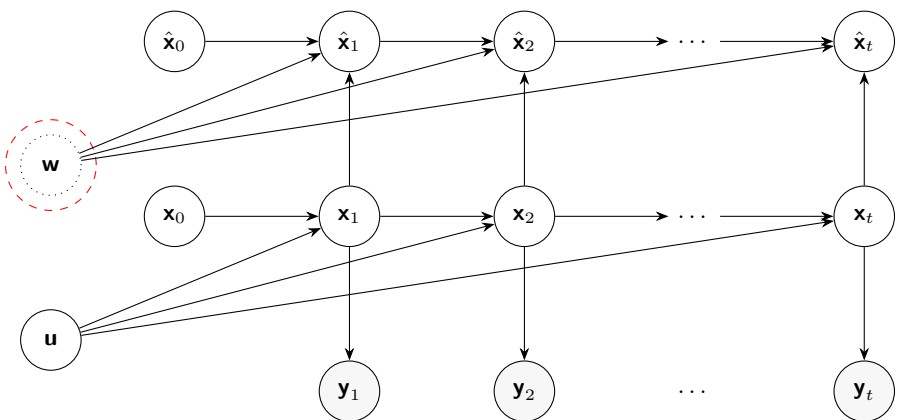

Figure 10: Directed graph of the generative model $\mathcal{M}_{\text{emu}}$ in an emulation problem, showing temporal dependencies ($t = 0, 1, 2, \ldots, T$). Observed variables $\mathbf{y}_t$ are shaded, and the global parameter $\mathbf{w}$ is highlighted.

Graphical model formalisms provide a general framework for estimating states and parameters in such systems. The key idea is that if we can establish a set of inference rules for arbitrary models, we can extend them to complex models like that in Figure 9.

### C.1 DIRECTED GRAPHS AND FACTOR GRAPHS

Starting from the structural equations defining a model, by sampling from the ancestral variable distributions and iteratively applying the generative model,

$$\mathbf{x} = \begin{bmatrix} \mathbf{x}_{\mathscr{O}_1} \\ \vdots \\ \mathbf{x}_{\mathscr{O}_J} \end{bmatrix} = \begin{bmatrix} \mathcal{P}_1(\mathbf{x}_{\mathscr{I}_1}) \\ \vdots \\ \mathcal{P}_J(\mathbf{x}_{\mathscr{I}_J}) \end{bmatrix}, \tag{25}$$

we obtain samples from the joint prior distribution.

For a valid structural equation model, the following must hold:

1. For all $j$, $\mathcal{O}_j \cap \mathcal{I}_j = \emptyset$ (i.e., no self-loops).

2. If $v \in \mathcal{O}_j$, then for all $k > j$, $v$ appears only in $\mathcal{I}_k$.

This defines a directed acyclic graph (DAG) over the variables, where nodes represent variables and edges represent generating equations. Each generating equation $j$ induces a density $p(\boldsymbol{x}_{\mathcal{O}_j} | \boldsymbol{x}_{\mathcal{I}_j})$, so that

$$p(\boldsymbol{x}) = \prod_j p(\boldsymbol{x}_{\mathcal{O}_j} | \boldsymbol{x}_{\mathcal{I}_j}). \tag{26}$$

Under mild conditions (local Markov, faithfulness, and consistency; see Koller & Friedman (2009)), this representation is equivalent to the joint distribution.

As a running example, consider:

$$\begin{bmatrix} \mathbf{x}_1 \\ \mathbf{x}_2 \\ \mathbf{x}_3 \\ \mathbf{x}_4 \\ \mathbf{x}_5 \\ \mathbf{x}_6 \end{bmatrix} = \begin{bmatrix} \mathcal{P}_1() \\ \mathcal{P}_{12}(\mathbf{x}_1) \\ \mathcal{P}_{13}(\mathbf{x}_1) \\ \mathcal{P}_{234}(\mathbf{x}_2, \mathbf{x}_3) \\ \mathcal{P}_{25}(\mathbf{x}_2) \\ \mathcal{P}_{456}(\mathbf{x}_4, \mathbf{x}_5) \end{bmatrix}. \tag{27}$$

This model is diagrammed in Figure 11a and its density factorizes as

$$p(\boldsymbol{x}) = p_1(\boldsymbol{x}_1) p_{12}(\boldsymbol{x}_2 | \boldsymbol{x}_1) p_{13}(\boldsymbol{x}_3 | \boldsymbol{x}_1) p_{234}(\boldsymbol{x}_4 | \boldsymbol{x}_2, \boldsymbol{x}_3) p_{25}(\boldsymbol{x}_5 | \boldsymbol{x}_2) p_{456}(\boldsymbol{x}_6 | \boldsymbol{x}_4, \boldsymbol{x}_5). \tag{28}$$

The directed generative model $\mathcal{M}$ is shown in Figure 11a.

The directed generative model is intuitive but not always convenient for inference. The factor graph representation (Frey et al., 1997; Kschischang et al., 2001) discards directional arrows and uses generic factors:

$$p(\boldsymbol{x}_1 | \boldsymbol{x}_2) \rightarrow f(\boldsymbol{x}_1, \boldsymbol{x}_2). \tag{29}$$

Thus, the running example density Equation 28 becomes

$$p(\boldsymbol{x}) = f_1(\boldsymbol{x}_1) f_{12}(\boldsymbol{x}_1, \boldsymbol{x}_2) f_{13}(\boldsymbol{x}_1, \boldsymbol{x}_3) f_{234}(\boldsymbol{x}_2, \boldsymbol{x}_3, \boldsymbol{x}_4) f_{25}(\boldsymbol{x}_2, \boldsymbol{x}_5) f_{456}(\boldsymbol{x}_4, \boldsymbol{x}_5, \boldsymbol{x}_6). \tag{30}$$

The corresponding factor graph is shown in Figure 11b. Factor graphs are bipartite, with nodes for both variables and factors, and they allow simple, local approximate belief-updating rules (introduced in Section C.2).

We now introduce the conditional graph transformation. In practice, we are interested in the evidence-conditional posterior $p(\boldsymbol{x}_{\mathcal{Q}} | \mathbf{x}_{\mathcal{E}} = \boldsymbol{x}_{\mathcal{E}}^*)$, i.e.,

$$p(\boldsymbol{x}_{\mathcal{Q}} | \mathbf{x}_{\mathcal{E}} = \boldsymbol{x}_{\mathcal{E}}^*) = \frac{p(\boldsymbol{x}_{\mathcal{Q}}, \boldsymbol{x}_{\mathcal{E}} = \boldsymbol{x}_{\mathcal{E}}^*)}{p(\boldsymbol{x}_{\mathcal{E}} = \boldsymbol{x}_{\mathcal{E}}^*)} \tag{31}$$

$$= \frac{\int p(\boldsymbol{x}_{\mathcal{Q}}, \boldsymbol{x}_{\mathcal{L}}, \boldsymbol{x}_{\mathcal{E}} = \boldsymbol{x}_{\mathcal{E}}^*) \, \mathrm{d}\boldsymbol{x}_{\mathcal{L}}}{\int p(\boldsymbol{x}_{\mathcal{Q}}, \boldsymbol{x}_{\mathcal{L}}, \boldsymbol{x}_{\mathcal{E}} = \boldsymbol{x}_{\mathcal{E}}^*) \, \mathrm{d}\boldsymbol{x}_{\mathcal{Q}} \, \mathrm{d}\boldsymbol{x}_{\mathcal{L}}}. \tag{32}$$

This conditional distribution can be represented as a factor graph, as shown in Figure 11c, where the observed nodes are fixed and the corresponding factors are replaced by conditional factors (denoted $f_{25}^*$ and $f_{456}^*$).

## C.2 FACTOR GRAPH GAUSSIAN BELIEF PROPAGATION

Loopy belief propagation (Murphy et al., 1999) is an efficient algorithm for approximate marginal inference in factor graphs. Although its convergence properties have been extensively studied (Wainwright & Jordan, 2008; Yedidia et al., 2005), we follow industrial practice (e.g. Dellaert & Kaess, 2017) and treat it as sufficiently accurate for our purposes.

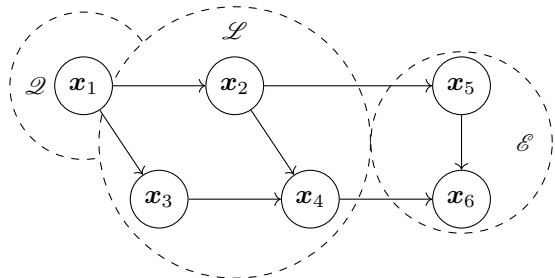

(a) Prior directed graph of generative model $\mathcal{M}$:
$p(\boldsymbol{x}) = p_1(\boldsymbol{x}_1)p_{12}(\boldsymbol{x}_2|\boldsymbol{x}_1)p_{13}(\boldsymbol{x}_3|\boldsymbol{x}_1)p_{25}(\boldsymbol{x}_5|\boldsymbol{x}_2)p_{234}(\boldsymbol{x}_4|\boldsymbol{x}_2,\boldsymbol{x}_3)p_{456}(\boldsymbol{x}_6|\boldsymbol{x}_4,\boldsymbol{x}_5).$

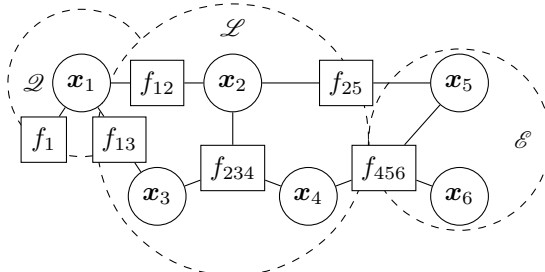

(b) Prior factor graph $\mathcal{G}$ for $\mathcal{M}$:
$p(\boldsymbol{x}) = f_1(\boldsymbol{x}_1)f_{12}(\boldsymbol{x}_1,\boldsymbol{x}_2)f_{13}(\boldsymbol{x}_1,\boldsymbol{x}_3)f_{234}(\boldsymbol{x}_2,\boldsymbol{x}_3,\boldsymbol{x}_4)f_{25}(\boldsymbol{x}_2,\boldsymbol{x}_5)f_{456}(\boldsymbol{x}_4,\boldsymbol{x}_5,\boldsymbol{x}_6).$

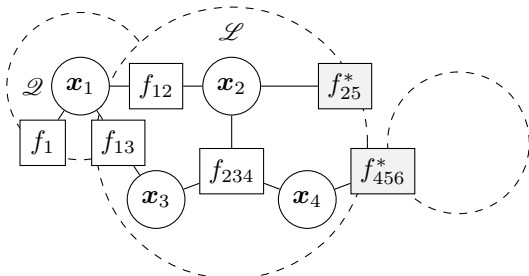

(c) Posterior factor graph $\mathcal{G}^*$ for $\mathcal{M}$ after observing $\boldsymbol{x}_5$ and $\boldsymbol{x}_6$, where
$p(\boldsymbol{x}_1,\boldsymbol{x}_2,\boldsymbol{x}_3,\boldsymbol{x}_4|\boldsymbol{x}_5,\boldsymbol{x}_6) \propto$
$f_1(\boldsymbol{x}_1)f_{12}(\boldsymbol{x}_1,\boldsymbol{x}_2)f_{13}(\boldsymbol{x}_1,\boldsymbol{x}_3)f_{234}(\boldsymbol{x}_2,\boldsymbol{x}_3,\boldsymbol{x}_4)f_{25}^*(\boldsymbol{x}_2)f_{456}^*(\boldsymbol{x}_4).$

Figure 11: Graphical model variants for the example model in Figure 11: (a) prior generative graph, (b) prior factor graph, and (c) posterior factor graph; with query nodes $\mathcal{Q} = \{1\}$, latent nodes $\mathcal{L} = \{2,3,4\}$, and observed nodes $\mathcal{E} = \{5,6\}$.

**Proposition 1** (Belief Propagation on Factor Graphs). *By iteratively propagating the messages,*

$$m_{f_j \to \mathbf{x}_k} = \int \left( f_j\left(\boldsymbol{x}_{\mathcal{N}_j}\right) \prod_{i \in \mathcal{N}_j \setminus k} m_{\mathbf{x}_i \to f_j} \right) \mathrm{d}\boldsymbol{x}_{\mathcal{N}_j \setminus k}, \tag{5}$$

$$m_{\mathbf{x}_k \to f_j} = \prod_{s \in \mathcal{N}_k \setminus j} m_{f_s \to \mathbf{x}_k}, \tag{6}$$

*BP approximates the marginals as*

$$b_{\mathcal{G}}\left(\boldsymbol{x}_k\right) = \prod_{s \in \mathcal{N}_k} m_{f_s \to \mathbf{x}_k} \approx \int p(\boldsymbol{x})\,\mathrm{d}\boldsymbol{x}_{\setminus k}. \tag{7}$$

The belief propagation algorithm for generic factor graphs, using the messages defined in Equations (Eq. 5), (Eq. 6), and (Eq. 7), is summarized in Algorithm 1 following the terminology of Ortiz et al. (2021).

---

**Algorithm 1** Loopy Low-rank Belief Propagation over Factor Graph $\mathcal{G}$

---

**Require:** Factor graph $\mathcal{G}$ with variable nodes $\{\mathbf{x}_k\}$ and factor nodes $\{f_j\}$
**Require:** Initial messages $\{\mathbf{x}_k \to f_j\}$ and $\{f_j \to \mathbf{x}_k\}$
**Ensure:** Approximate marginal beliefs $b(\mathbf{x}_k)$ for all $\mathbf{x}_k \in \mathcal{G}$
 1: Initialize message queues $Q_{\mathbf{x}_k \to f_j}$ and $Q_{f_j \to \mathbf{x}_k}$ as empty.
 2: **while** not converged **do**
 3:    **for** each factor $f_j \in \mathcal{G}$ **do**
 4:       **for** each variable $\mathbf{x}_k \in \mathcal{N}_{f_j}$ **do**
 5:          Send factor-to-variable message (cf. (Eq. 5)).
 6:       **end for**
 7:    **end for**
 8:    **for** each variable $\mathbf{x}_k \in \mathcal{G}$ **do**
 9:       **for** each factor $f_j \in \mathcal{N}_{\mathbf{x}_k}$ **do**
10:          Send variable-to-factor message (cf. (Eq. 6)).
11:       **end for**
12:       Update belief $b(\boldsymbol{x}_k)$ using (Eq. 7).
13:    **end for**
14:    Check convergence criteria.
15: **end while**
16: **return** $b(\mathbf{x}_k)$ for all $\mathbf{x}_k \in \mathcal{G}$.

---

### C.3 GAUSSIAN FACTOR UPDATES

The factor graph formalism is especially convenient for Gaussian models because the Gaussian density is closed under multiplication, conditioning, and marginalisation. Although we introduced these ideas in Section 2.2.3, we detail them here.

Working with Gaussians in canonical form is advantageous (Eustice et al., 2006). We denote by $\phi_M(\boldsymbol{x}; \boldsymbol{m}, \mathrm{K})$ the Gaussian density with mean $\boldsymbol{m}$ and covariance $\mathrm{K}$.

**Proposition 5.** *Partition the random vector as*

$$\mathbf{x}_j \sim \phi_M\left(\begin{bmatrix}\boldsymbol{x}_k \\ \boldsymbol{x}_\ell\end{bmatrix}; \begin{bmatrix}\boldsymbol{m}_k \\ \boldsymbol{m}_\ell\end{bmatrix}, \begin{bmatrix}\mathrm{K}_{kk} & \mathrm{K}_{k\ell} \\ \mathrm{K}_{\ell k} & \mathrm{K}_{\ell\ell}\end{bmatrix}\right). \tag{33}$$

*Define $\hat{\mathrm{K}} := \left(\mathrm{K}_{jj}^{-1} + \mathrm{K}_{jj}'^{-1}\right)^{-1}$. Then, for Gaussian nodes and factors, the operations in Definition 1 take the following forms.*

*Conditioning:*

$$\phi_M\left(\boldsymbol{x}_j; \boldsymbol{m}_j, \mathrm{K}_{jj}\right), \quad \boldsymbol{x}_k^* \mapsto \phi_M\left(\boldsymbol{x}_\ell; \boldsymbol{m}_\ell + \mathrm{K}_{\ell k}\mathrm{K}_{kk}^{-1}(\boldsymbol{x}_k^* - \boldsymbol{m}_k), \mathrm{K}_{\ell\ell} - \mathrm{K}_{\ell k}\mathrm{K}_{kk}^{-1}\mathrm{K}_{k\ell}\right) \tag{34}$$

*Marginalisation:*

$$\phi_M\left(\boldsymbol{x}_j; \boldsymbol{m}_j, \mathrm{K}_{jj}\right) \mapsto \phi_M\left(\boldsymbol{x}_k; \boldsymbol{m}_k, \mathrm{K}_{kk}\right) \tag{35}$$

*Multiplication (moments):*

$$\phi_M'\left(\boldsymbol{x}_j; \boldsymbol{m}_j', \mathrm{K}_{jj}'\right), \quad \phi_M\left(\boldsymbol{x}_j; \boldsymbol{m}_j, \mathrm{K}_{jj}\right) \mapsto \phi_M\left(\boldsymbol{x}_j; \hat{\mathrm{K}}\left(\mathrm{K}_{jj}^{-1}\boldsymbol{m}_j + \mathrm{K}_{jj}'^{-1}\boldsymbol{m}_j'\right), \hat{\mathrm{K}}\right) \tag{36}$$

*Multiplication (canonical):*

$$\phi_C'\left(\boldsymbol{x}_j; \boldsymbol{n}_j', \mathrm{P}_{jj}'\right), \quad \phi_C\left(\boldsymbol{x}_j; \boldsymbol{n}_j, \mathrm{P}_{jj}\right) \mapsto \phi_C\left(\boldsymbol{x}_j; \boldsymbol{n}_j' + \boldsymbol{n}_j, \mathrm{P}_{jj}' + \mathrm{P}_{jj}\right) \tag{37}$$

*Proof:* See Bickson (2009). $\quad\square$

Note that Equation 37 is Equation 14 in the main text.

### C.4 GAUSSIAN APPROXIMATION OF NON-GAUSSIAN FACTOR POTENTIALS

For simplicity, assume that all factor potentials are bivariate with $\mathbf{x}_2 = \mathcal{P}(\mathbf{x}_1)$. (If not, the variates can be stacked and relabeled accordingly.)

A classic approach to approximating the factor potential generated by a nonlinear process is the propagation-of-errors, also known as the $\delta$-method (Dorfman, 1938). The $\delta$-method uses a first-order Taylor expansion to approximate how a nonlinear function transforms the moments (e.g., the mean and covariance) of a random variable. This is analogous to a Laplace approximation but is specifically used for error propagation in moments. In its simplest form, it approximates

$$\mathbb{E}f(\mathbf{x}_1) \approx f\left(\mathbb{E}\mathbf{x}_1\right), \tag{38}$$

which is justified by Taylor expansion.

To estimate the joint covariance of the transformed variable, define

$$f(\boldsymbol{x}) := \begin{bmatrix} \boldsymbol{x} \\ \mathcal{P}(\boldsymbol{x}) \end{bmatrix} \begin{bmatrix} \boldsymbol{x} \\ \mathcal{P}(\boldsymbol{x}) \end{bmatrix}^{\top} - \mathbb{E}\left[\begin{bmatrix} \boldsymbol{x} \\ \mathcal{P}(\boldsymbol{x}) \end{bmatrix}\right] \mathbb{E}\left[\begin{bmatrix} \boldsymbol{x} \\ \mathcal{P}(\boldsymbol{x}) \end{bmatrix}\right]^{\top}. \tag{39}$$

The difference

$$\mathcal{E}_{\text{Jensen}}(\mathbf{x}_1, f) := \mathbb{E}f(\mathbf{x}_1) - f\left(\mathbb{E}\mathbf{x}_1\right) \tag{40}$$

is called the *Jensen gap* and represents one source of error in the linearisation.

Since the function $\mathcal{P}$ is typically nonlinear and intractable for an exact moment calculation, the $\delta$-method further approximates $f$ using a first-order Taylor expansion about the mean:

$$\hat{f}(\boldsymbol{x}) \approx f\left(\mathbb{E}\mathbf{x}_1\right) + \nabla_{\boldsymbol{x}'} f(\boldsymbol{x}')|_{\boldsymbol{x}'=\mathbb{E}\mathbf{x}_1} (\boldsymbol{x} - \mathbb{E}\mathbf{x}_1). \tag{41}$$

The error in this Taylor approximation is defined as

$$\mathcal{E}_{\text{Taylor}}(\mathbf{x}_1, f, \hat{f}) := \mathbb{E}f(\mathbf{x}_1) - \mathbb{E}[\hat{f}(\mathbf{x}_1)]. \tag{42}$$

In summary, the overall error in approximating the transformed covariance using the $\delta$-method includes the Jensen gap, the Taylor error, and any additional error due to non-Gaussianity of the inputs. Although the $\delta$-method is simple and computationally efficient, it is not always optimal in terms of accuracy; alternative methods (such as higher-order approximations or sigma-point approaches, or ensemble approximations) might sometimes yield better results.

## D MINIMUM VIABLE INTRODUCTION TO THE ENSEMBLE KALMAN FILTER

The field of Ensemble Kalman methods is mature and extensive; see, e.g., Evensen (2009) for a comprehensive treatment. Here we provide a brief introduction sufficient for our purposes.

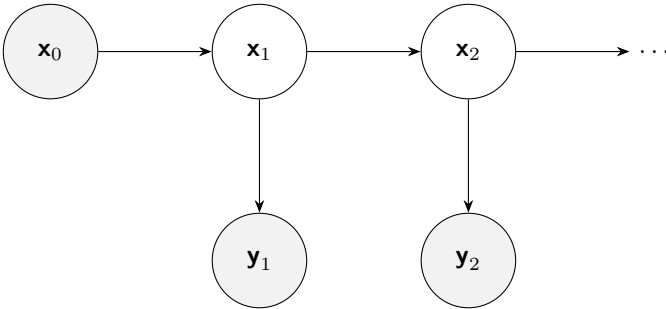

Figure 12: Generative model for a state filtering problem, where hidden states $\mathbf{x}_t$ evolve over time and produce observations $\mathbf{y}_t$ for $t = 1, 2, \ldots$.

The central idea is as follows: Given the state-space model in Figure 12, where the joint prior density factors as

$$p(\boldsymbol{x}_{0:T}, \boldsymbol{y}_{1:T}) = \prod_{t=1}^{T} p(\boldsymbol{x}_t | \boldsymbol{x}_{t-1}) \, p(\boldsymbol{y}_t | \boldsymbol{x}_t),$$

filtering seeks the conditional density $p(\boldsymbol{x}_{0:T} | \boldsymbol{y}_{1:T})$. By induction, if we know an estimate of $p(\boldsymbol{x}_{0:T-1} | \boldsymbol{y}_{1:T-1})$, then incorporating the new observation $\boldsymbol{y}_T$ only requires multiplying by $p(\boldsymbol{x}_T | \boldsymbol{x}_{T-1}) \, p(\boldsymbol{y}_T | \boldsymbol{x}_T)$ and normalizing.

Under the assumptions of linearity and Gaussianity, let

$$p(\boldsymbol{x}_{T-1} | \boldsymbol{y}_{1:T-1}) = \phi_M(\boldsymbol{x}_{T-1}; \boldsymbol{m}_{T-1}, \mathrm{K}_{T-1}).$$

Linearity implies that the state transition can be written as

$$\boldsymbol{x}_T = \mathcal{P}_{T-1}(\boldsymbol{x}_{T-1}) = \mathrm{F}_{T-1}\boldsymbol{x}_{T-1} + \boldsymbol{d}_{T-1} + \boldsymbol{\epsilon}_T, \tag{43}$$

with $\boldsymbol{\epsilon}_T \sim \mathcal{N}(\boldsymbol{0}, \mathrm{Q}_T)$. Consequently, the joint density for $\boldsymbol{x}_{T-1}$ and $\boldsymbol{x}_T$ (conditioned on $\boldsymbol{y}_{1:T-1}$) is

$$\phi_M\left( \begin{bmatrix} \boldsymbol{x}_{T-1} \\ \boldsymbol{x}_T \end{bmatrix}; \begin{bmatrix} \boldsymbol{m}_{T-1} \\ \boldsymbol{d}_{T-1} + \mathrm{F}_{T-1}\boldsymbol{m}_{T-1} \end{bmatrix}, \begin{bmatrix} \mathrm{K}_{T-1} & \mathrm{K}_{T-1}\mathrm{F}_{T-1}^\top \\ \mathrm{F}_{T-1}\mathrm{K}_{T-1} & \mathrm{F}_{T-1}\mathrm{K}_{T-1}\mathrm{F}_{T-1}^\top + \mathrm{Q}_T \end{bmatrix} \right).$$

The marginal for $\boldsymbol{x}_T$ is then

$$p(\boldsymbol{x}_T | \boldsymbol{y}_{1:T-1}) = \phi_M(\boldsymbol{x}_T; \tilde{\boldsymbol{m}}_T, \tilde{\mathrm{K}}_T), \tag{44}$$

with

$$\tilde{\boldsymbol{m}}_T = \mathrm{F}_{T-1}\boldsymbol{m}_{T-1} + \boldsymbol{d}_{T-1}, \quad \tilde{\mathrm{K}}_T = \mathrm{F}_{T-1}\mathrm{K}_{T-1}\mathrm{F}_{T-1}^\top + \mathrm{Q}_T.$$

Next, conditioning on the observation $\boldsymbol{y}_T$ (assuming the linear observation model $\boldsymbol{y}_T = \mathrm{H}_T\boldsymbol{x}_T + \boldsymbol{e}_T + \varepsilon_T$ with $\varepsilon_T \sim \mathcal{N}(\boldsymbol{0}, \mathrm{R}_T)$) yields the standard Kalman filter update:

$$\mathrm{K}_T = \tilde{\mathrm{K}}_T - \tilde{\mathrm{K}}_T \mathrm{H}_T^\top \left( \mathrm{H}_T \tilde{\mathrm{K}}_T \mathrm{H}_T^\top + \mathrm{R}_T \right)^{-1} \mathrm{H}_T \tilde{\mathrm{K}}_T, \tag{45}$$

$$\boldsymbol{m}_T = \tilde{\boldsymbol{m}}_T + \mathrm{K}_T \mathrm{H}_T^\top \left( \mathrm{H}_T \tilde{\mathrm{K}}_T \mathrm{H}_T^\top + \mathrm{R}_T \right)^{-1} \left( \boldsymbol{y}_T - \mathrm{H}_T \tilde{\boldsymbol{m}}_T - \boldsymbol{e}_T \right). \tag{46}$$

In practice, state transitions or observation models may be nonlinear. The Extended Kalman Filter (EKF) approximates these models by linearizing via Taylor expansion (propagation of errors). For example, one sets

$$\boldsymbol{d}_{T-1} \approx \mathbb{E}[\mathcal{P}_{T-1}(\boldsymbol{m}_{T-1})], \quad \mathrm{F}_{T-1} \approx \nabla_{\boldsymbol{x}} \mathcal{P}_{T-1}\Big|_{\boldsymbol{x}=\boldsymbol{m}_{T-1}},$$

and similarly for the observation model, while choosing diagonal covariance matrices $\mathrm{Q}_T$ and $\mathrm{R}_T$. However, the EnKF takes a different approach.

Rather than computing updates in closed form, the Ensemble Kalman Filter (EnKF) approximates the filtering distribution using an ensemble of Monte Carlo samples. Let

$$\mathrm{X} = \begin{bmatrix} \boldsymbol{x}^{(1)} & \cdots & \boldsymbol{x}^{(N)} \end{bmatrix}$$

be a matrix of $N$ samples drawn from $p(\boldsymbol{x}; \boldsymbol{m}, \mathrm{K})$. The empirical ensemble statistics are defined as follows:

$$\overline{\mathrm{X}} := \mathrm{X}\mathrm{A}_N, \quad \text{where } \mathrm{A}_N := \frac{1}{N}\mathbf{1}_{N \times 1}, \tag{47}$$

$$\check{\mathrm{X}} := \mathrm{X} - \overline{\mathrm{X}}\mathrm{B}_N, \quad \text{with } \mathrm{B}_N := \mathbf{1}_{1 \times N},$$

$$\widehat{\mathbb{E}}\mathrm{X} = \overline{\mathrm{X}}, \tag{48}$$

$$\widehat{\mathrm{Var}}_\mathrm{V}\mathrm{X} = \frac{1}{N-1}\check{\mathrm{X}}\check{\mathrm{X}}^\top + \mathrm{V}, \tag{49}$$

$$\widehat{\mathrm{Cov}}(\mathrm{X}, \mathrm{Y}) = \frac{1}{N-1}\check{\mathrm{X}}\check{\mathrm{Y}}^\top, \tag{50}$$

where V is a diagonal "nugget" term (often set to $\sigma^2 I$) to ensure numerical stability.

The EnKF propagates the ensemble forward using the model:

$$
\begin{bmatrix} X_{T-1} \\ X_T \end{bmatrix} \Big| \boldsymbol{y}_{1:T-1} := \begin{bmatrix} X_{T-1} \Big| \boldsymbol{y}_{1:T-1} \\ \mathcal{P}(X_{T-1} \Big| \boldsymbol{y}_{1:T-1}) \end{bmatrix}.
\tag{51}
$$

Thus, the EnKF approximates the joint density by the empirical distribution defined by the ensemble without ever explicitly evaluating the Gaussian density.

A key step is the ensemble update (often referred to as the Matheron update).

**Proposition 2.** *Partition* $\mathbf{x}^\top = \begin{bmatrix} \mathbf{x}_k^\top & \mathbf{x}_\ell^\top \end{bmatrix}$ *such that* $X^\top = \begin{bmatrix} X_k^\top & X_\ell^\top \end{bmatrix}$. *Assume the ensemble* X *follows the Gaussian distribution:*

$$
X \sim \phi_M \left( \begin{bmatrix} \boldsymbol{x}_k \\ \boldsymbol{x}_\ell \end{bmatrix} ; \begin{bmatrix} \overline{X}_k \\ \overline{X}_\ell \end{bmatrix}, \begin{bmatrix} \widehat{\mathrm{Var}}_V X_k & \widehat{\mathrm{Cov}}(X_\ell, X_k) \\ \widehat{\mathrm{Cov}}(X_k, X_\ell) & \widehat{\mathrm{Var}}_V(X_\ell) \end{bmatrix} \right).
\tag{16}
$$

*In ensemble form, conditioning (Eq. 9) is performed as:*

$$
X, \boldsymbol{x}_k^* \mapsto X_\ell + \widehat{\mathrm{Cov}}(X_\ell, X_k) \widehat{\mathrm{Var}}_V^{-1}(X_k)(\boldsymbol{x}_k^* B - X_k)
\tag{17}
$$

*The computational cost of solving Equation 17 is* $\mathcal{O}(N^3 + N^2 D_{\mathbf{x}_k})$. *Marginalization (Eq. 10) is simply truncation, i.e.,* $X \mapsto X_k$.

*Proof:* The equality Equation 17 follows from 6 with the substitution of Equation 16. which can be computed with a cost of $\mathcal{O}(N^3 + N^2 D_{\mathbf{x}_k})$ using the Woodbury formula to efficiently solve the linear system involving $\widehat{\mathrm{Var}}_V^{-1}(X_k)$ (MacKinlay, 2025). Marginalization is simply performed by truncating the ensemble to the relevant components. $\square$

We can use Equation 17 to calculate $X_T | \boldsymbol{y}_{1:T}$ by using $\boldsymbol{y}_T$ as the observations $\boldsymbol{x}_k^*$ in the update. We have thus obtained sampled from the filtered distribution without ever evaluating its density. This update is then used to condition the ensemble on new observations. For example, if the joint ensemble of states and observations is

$$
\begin{bmatrix} X_T \\ Y_T \end{bmatrix} \Big| \boldsymbol{y}_{1:T-1} = \begin{bmatrix} X_T \Big| \boldsymbol{y}_{1:T-1} \\ \mathcal{H}(X_T \Big| \boldsymbol{y}_{1:T-1}) \end{bmatrix},
$$

one uses Equation 17 with the observed $\boldsymbol{y}_T$ in place of $\boldsymbol{x}_k^*$ to obtain the updated ensemble $X_T | \boldsymbol{y}_{1:T}$. In this way, samples from the filtered distribution are obtained without explicit density evaluations.

### D.1 Gaussian approximation of non-Gaussian factor potentials

Unlike the GaBP, which approximates non-Gaussian factors using the $\delta$-method (i.e. via Taylor expansion and linearisation), the EnKF avoids this entirely by using empirical samples to estimate the moments of the joint distribution. Although this stochastic approximation introduces additional (aleatoric) noise, empirical studies (e.g. Furrer & Bengtsson (2007)) suggest that the EnKF often produces more accurate estimates than the Gaussian approximations used in GaBP. In particular, by sampling from the full joint distribution, the EnKF bypasses the Jensen gap and Taylor errors inherent to the $\delta$-method.

Heuristically, this means that while the GaBP may produce an approximate joint variance of the form

$$
\widehat{\mathrm{Var}}^{\mathrm{GaBP}} \begin{bmatrix} \mathbf{x}_1 \\ \mathbf{x}_2 \end{bmatrix} \simeq \begin{bmatrix} \mathrm{Var}(\mathbf{x}_1) & J(\boldsymbol{m}_1)\,\mathrm{Var}(\mathbf{x}_1) \\ \mathrm{Var}(\mathbf{x}_1)\,J(\boldsymbol{m}_1)^\top & J(\boldsymbol{m}_1)\,\mathrm{Var}(\mathbf{x}_1)\,J^\top(\boldsymbol{m}_1) \end{bmatrix},
\tag{52}
$$

with $J(\boldsymbol{m}_1)$ the Jacobian of $\mathcal{P}$ at the mean, the EnKF estimates the joint variance empirically as

$$
\widehat{\mathrm{Var}}^{\mathrm{EnKF}} \begin{bmatrix} \mathbf{x}_1 \\ \mathbf{x}_2 \end{bmatrix} \simeq \widehat{\mathrm{Var}}_{\sigma^2 I} \begin{bmatrix} X^{(1)} & \cdots & X^{(N)} \\ \mathcal{P}(X^{(1)}) & \cdots & \mathcal{P}(X^{(N)}) \end{bmatrix},
\tag{53}
$$

where the latter includes a nugget term to maintain numerical stability. Although the relative quality of these approximations is complex to characterize theoretically, empirical evidence supports the superior performance of the EnKF approach in many high-dimensional problems.

## E  MATHERON UPDATES FOR GAUSSIAN VARIATES

The pathwise update for Gaussian processes, known as the *Matheron update* (see, e.g., Wilson et al. (2021); Doucet (2010)), provides a method to generate samples from a conditional distribution without directly computing its density. Suppose

$$\begin{bmatrix} \mathbf{y} \\ \mathbf{w} \end{bmatrix} \sim \mathcal{N}\left( \begin{bmatrix} m_{\mathbf{y}} \\ m_{\mathbf{w}} \end{bmatrix}, \begin{bmatrix} \mathrm{K}_{\mathbf{yy}} & \mathrm{K}_{\mathbf{yw}} \\ \mathrm{K}_{\mathbf{wy}} & \mathrm{K}_{\mathbf{ww}} \end{bmatrix} \right). \tag{54}$$

Then the conditional moments are given by

$$\mathbb{E}[\mathbf{y} \mid \mathbf{w} = \boldsymbol{w}^*] = m_{\mathbf{y}} + \mathrm{K}_{\mathbf{wy}}\,\mathrm{K}_{\mathbf{ww}}^{-1}(\boldsymbol{w}^* - m_{\mathbf{w}}), \tag{55}$$

$$\mathrm{Var}[\mathbf{y} \mid \mathbf{w} = \boldsymbol{w}^*] = \mathrm{K}_{\mathbf{yy}} - \mathrm{K}_{\mathbf{wy}}\,\mathrm{K}_{\mathbf{ww}}^{-1}\mathrm{K}_{\mathbf{yw}}. \tag{56}$$

**Proposition 6.** *For* $\begin{bmatrix} \mathbf{y} \\ \mathbf{w} \end{bmatrix}$ *as in Equation 54, the mapping*

$$\mathbf{y}^* := \mathbf{y} + \mathrm{K}_{\mathbf{wy}}\,\mathrm{K}_{\mathbf{ww}}^{-1}(\boldsymbol{w}^* - \mathbf{w}) \tag{57}$$

*produces a random variable* $\mathbf{y}^*$ *that has the same distribution as* $\mathbf{y}$ *conditioned on* $\mathbf{w} = \boldsymbol{w}^*$, *i.e.,* $\mathbf{y}^* \stackrel{\mathrm{d}}{=} (\mathbf{y} \mid \mathbf{w} = \boldsymbol{w}^*)$.

*Proof:* Taking the expectation of Equation 57 gives

$$\mathbb{E}[\mathbf{y}^*] = m_{\mathbf{y}} + \mathrm{K}_{\mathbf{wy}}\,\mathrm{K}_{\mathbf{ww}}^{-1}(\boldsymbol{w}^* - m_{\mathbf{w}}),$$

which matches Equation 55. For the variance, since $\boldsymbol{w}^*$ is fixed and $\mathbf{w}$ is Gaussian, one can show that

$$\mathrm{Var}[\mathbf{y}^*] = \mathrm{K}_{\mathbf{yy}} - \mathrm{K}_{\mathbf{wy}}\,\mathrm{K}_{\mathbf{ww}}^{-1}\mathrm{K}_{\mathbf{yw}},$$

which is Equation 56. Thus, the mapping in Equation 57 indeed produces samples with the desired conditional moments. $\square$

Note that this update does not require explicit evaluation of the density of $\mathbf{y}$ and forms the basis for pathwise (sample-based) updates in Gaussian process models.

## F  DIAGONAL MATRICES WITH LOW-RANK PERTURBATION

Suppose

$$\mathrm{K} = \mathrm{V} + s\,\mathrm{L}\mathrm{L}^{\top},$$

where L is a $D \times N$ matrix, $\mathrm{V} = \mathrm{diag}(\boldsymbol{v})$ (with $\boldsymbol{v}$ a $D$-vector), and

$$s \in \{-1, 1\}$$

specifies the sign of the low-rank update. We are primarily interested in such matrices when $N \ll D$; in that case we call them *Diagonal Matrix with Low-rank perturbation* because their computational properties are favourable. (In contrast, matrices without such exploitable structure we refer to as *dense*.)

Throughout this section we assume that the matrices in question are positive definite and that all dimensions are conformable. We refer to the matrix L as the *component* of the DLR matrix, and the diagonal matrix V as the *nugget* term (by analogy with classic kriging). Note that if $N \geq D$ the name is misleading since the matrix is not truly low-rank; the identities below still hold but the computational advantages are lost.

### F.1  MULTIPLICATION BY A DENSE MATRIX

The product of an arbitrary dense matrix X with a DLR matrix can be computed efficiently by grouping operations. In particular, if

$$\mathrm{K} = \mathrm{V} + s\,\mathrm{L}\mathrm{L}^{\top},$$

then

$$\mathrm{K}\mathrm{X} = \mathrm{V}\mathrm{X} + s\,\mathrm{L}\left(\mathrm{L}^{\top}\mathrm{X}\right).$$

This requires only $\mathcal{O}(D \cdot \dim(X))$ operations for the diagonal term and $\mathcal{O}(DN\dim(X))$ for the low-rank term, and it can be computed without forming the full dense matrix K. (In general the product is not itself a DLR matrix.)

## F.2 Addition

The sum of two DLR matrices with the same sign is also a DLR matrix. Suppose

$$\mathrm{K} = \mathrm{V} + s\,\mathrm{LL}^\top, \quad \mathrm{K}' = \mathrm{V}' + s\,\mathrm{L}'\mathrm{L}'^\top.$$

Then

$$\mathrm{K} + \mathrm{K}' = \mathrm{V} + \mathrm{V}' + s\left(\mathrm{LL}^\top + \mathrm{L}'\mathrm{L}'^\top\right) \tag{58}$$

$$= \left(\mathrm{V} + \mathrm{V}'\right) + s\,[\mathrm{L} \quad \mathrm{L}']\,[\mathrm{L} \quad \mathrm{L}']^\top. \tag{59}$$

Thus the sum is a DLR matrix with nugget $\mathrm{V} + \mathrm{V}'$ and component $[\mathrm{L} \quad \mathrm{L}']$.

## F.3 DLR inverses of DLR matrices

Inverses of DLR matrices can themselves be written in DLR form and computed efficiently.

**Proposition 7.** *Let*

$$\mathrm{K} = \mathrm{V} + \mathrm{LL}^\top$$

*be a DLR matrix with positive sign ($s = +1$). Then its inverse is*

$$\mathrm{K}^{-1} = \mathrm{V}^{-1} - \mathrm{RR}^\top, \tag{60}$$

*where*

$$\mathrm{R} = \mathrm{V}^{-1}\mathrm{L}\,\mathrm{chol}\!\left(\left(\mathrm{I} + \mathrm{L}^\top\mathrm{V}^{-1}\mathrm{L}\right)^{-1}\right).$$

*Here* $\mathrm{chol}(\mathrm{A})$ *denotes a (conventional) Cholesky factor satisfying* $\mathrm{chol}(\mathrm{A})\,\mathrm{chol}(\mathrm{A})^\top = \mathrm{A}$.

*Proof:* The Woodbury identity gives

$$\mathrm{K}^{-1} = \mathrm{V}^{-1} - \mathrm{V}^{-1}\mathrm{L}\left(\mathrm{I} + \mathrm{L}^\top\mathrm{V}^{-1}\mathrm{L}\right)^{-1}\mathrm{L}^\top\mathrm{V}^{-1}.$$

Defining $\mathrm{R}$ as above yields Equation 60. $\qquad\square$

**Proposition 8.** *Let*

$$\mathrm{P} = \mathrm{U} - \mathrm{RR}^\top$$

*be a DLR matrix with negative sign ($s = -1$). Then its inverse is given by*

$$\mathrm{P}^{-1} = \mathrm{U}^{-1} + \mathrm{LL}^\top, \tag{61}$$

*where*

$$\mathrm{L} = \mathrm{U}^{-1}\mathrm{R}\,\mathrm{chol}\!\left(\left(\mathrm{I} - \mathrm{R}^\top\mathrm{U}^{-1}\mathrm{R}\right)^{-1}\right).$$

*Proof:* Applying the Woodbury identity for a negative low-rank update, we have

$$\left(\mathrm{U} - \mathrm{RR}^\top\right)^{-1} = \mathrm{U}^{-1} + \mathrm{U}^{-1}\mathrm{R}\left(\mathrm{I} - \mathrm{R}^\top\mathrm{U}^{-1}\mathrm{R}\right)^{-1}\mathrm{R}^\top\mathrm{U}^{-1}.$$

Defining $\mathrm{L}$ as above, we obtain Equation 61. $\qquad\square$

The computational cost of either inversion is $\mathcal{O}(N^2 D + N^3)$: roughly $\mathcal{O}(N^3)$ for constructing the Cholesky factor and $\mathcal{O}(DN^2)$ for the necessary matrix multiplications. The space cost is $\mathcal{O}(ND)$. For a given $\mathrm{K} = \mathrm{V} + s\,\mathrm{LL}^\top$, the term

$$s\,\mathrm{I} + \mathrm{L}^\top\mathrm{V}^{-1}\mathrm{L}$$

is conventionally referred to as the *capacitance* of the matrix.

### F.4 EXACT INVERSION WHEN THE COMPONENT IS HIGH-RANK

Suppose instead that the low-rank component is not low-rank at all (i.e. the $D \times N$ component satisfies $N > D$). In this case the cost of computing the low-rank inversion—dominated by a $\mathcal{O}(N^3)$ Cholesky factorisation—exceeds the cost of naively inverting the dense $D \times D$ matrix (which is $\mathcal{O}(D^3)$). In such circumstances it can be preferable to compute the full dense inverse and then recover a DLR representation via spectral decomposition.

For example, if one computes $\mathrm{P}^{-1}$ (with P as in Proposition 8), then since

$$\mathrm{P}^{-1} = \mathrm{U}^{-1} + \mathrm{LL}^\top,$$

one may obtain a low-rank factor by forming the spectral decomposition

$$\mathrm{P}^{-1} - \mathrm{U}^{-1} = \mathrm{Q}\Lambda\mathrm{Q}^\top$$

and setting

$$\mathrm{L} = \mathrm{Q}\,\Lambda^{1/2}.$$

(In the special case where U is a constant diagonal, more efficient methods may be available.)

A similar strategy applies for inverting a DLR matrix of the form

$$\mathrm{K} = \mathrm{V} + \mathrm{LL}^\top,$$

by computing its dense inverse $\mathrm{K}^{-1}$ and then recovering the low-rank representation from the identity

$$\mathrm{K}^{-1} = \mathrm{V}^{-1} - \mathrm{RR}^\top.$$

### F.5 REDUCING COMPONENT RANK

We can use the singular value decomposition (SVD) to reduce the rank of the component in a DLR matrix in a way that minimizes the Frobenius norm error. Suppose

$$\mathrm{K} = \mathrm{V} + \mathrm{LL}^\top,$$

where L is a $D \times M$ matrix and $\mathrm{V} = \mathrm{diag}(\boldsymbol{v})$. Let the thin SVD of L be

$$\mathrm{L} = \mathrm{YSZ}^\top,$$

with $\mathrm{Y} \in \mathbb{R}^{D \times M}$, $\mathrm{S} \in \mathbb{R}^{M \times M}$ (diagonal with nonnegative entries), and $\mathrm{Z} \in \mathbb{R}^{M \times M}$, so that $\mathrm{Y}^\top\mathrm{Y} = \mathrm{I}_M$ and $\mathrm{Z}^\top\mathrm{Z} = \mathrm{I}_M$. Then

$$\mathrm{LL}^\top = \mathrm{YS}^2\mathrm{Y}^\top.$$

Any zero singular values can be removed exactly. Moreover, if we retain only the largest $r$ singular values (setting the others to zero), we obtain the best Frobenius-norm approximation of $\mathrm{LL}^\top$ of rank $r$. (A conventional thin SVD of L costs $\mathcal{O}(DM^2)$; if $M$ is large one may consider randomized methods.)

## G GAUSSIANS WITH DLR PARAMETERS

We recall the two common parameterisations of the multivariate Gaussian used in section Section 2.2.3. In *moments* form, we write

$$\mathbf{x} \sim \mathcal{N}(\boldsymbol{m}, \mathrm{K}),$$

where $\boldsymbol{m} = \mathbb{E}[\mathbf{x}]$ and $\mathrm{K} = \mathrm{Var}(\mathbf{x})$. In *canonical* form the same density is written as

$$\mathbf{x} \sim \mathcal{N}_C(\boldsymbol{n}, \mathrm{P}),$$

where

$$\boldsymbol{n} = \mathrm{K}^{-1}\boldsymbol{m}, \quad \mathrm{P} = \mathrm{K}^{-1}.$$

(Up to normalizing constants the exponent in the density may be written either as $-\frac{1}{2}(\boldsymbol{x} - \boldsymbol{m})^\top \mathrm{K}^{-1}(\boldsymbol{x} - \boldsymbol{m})$ or equivalently as $-\frac{1}{2}\boldsymbol{x}^\top \mathrm{P}\boldsymbol{x} + \boldsymbol{n}^\top\boldsymbol{x}$.)

A given Gaussian ensemble is naturally associated with a moments-form Gaussian,

$$\mathrm{X}_{\sigma^2} \sim \mathcal{N}(\boldsymbol{m}, \mathrm{K}),$$

where we write

$$\boldsymbol{m} = \widehat{\mathbb{E}}[X], \qquad K = \widehat{\mathrm{Var}}_{\sigma^2}(X) = \sigma^2 I + \breve{X}\breve{X}^\top,$$

introducing a parameter $\sigma^2$ to ensure invertibility of the covariance if needed, or more generally substituting an arbitrary diagonal matrix for $\sigma^2 I$.

By applying Proposition 7 to the DLR form of the covariance, one may compute the canonical parameters cheaply. That is, writing

$$K^{-1} = \widehat{\mathrm{Var}}_{\sigma^2}^{-1}(X) \equiv P = \sigma^{-2}I - RR^\top,$$

we have

$$\boldsymbol{n} = P\,\boldsymbol{m}.$$

Conversely, given $\boldsymbol{n}$ and $P = \sigma^{-2}I - RR^\top$ one may recover $\boldsymbol{m}$ and K via

$$K = P^{-1} = \sigma^{-2}I - RR^\top, \qquad \boldsymbol{m} = P^{-1}\boldsymbol{n} = \sigma^{-2}\boldsymbol{n} - RR^\top\boldsymbol{n},$$

using the alternative low-rank inverse formula (see Proposition 8).

# H  GENBP DETAILS

## H.1  GENBP ALGORITHM

Here we detail the GEnBP algorithm, specifying the matrix operations needed to perform the Gaussian updates while maintaining the DLR forms for the matrix parameters.

---

**Algorithm 2:** GEnBP

---

**Require:** Graph $\mathcal{G}$, generative processes $\{\mathcal{P}_j\}_j$, observations $\boldsymbol{x}_{\mathscr{E}}$, and an ancestral ensemble sample $X_{\mathscr{A}}$.
**Ensure:** An observation-conditional sample $X_{\mathscr{Q}}^*$.
 1: **while** not converged **do**
 2:     Sample an ensemble from the generative processes $\mathcal{P}_j$ on $\mathcal{G}$ (using (Eq. 3)).
 3:     Convert $\mathcal{G}$ into the conditional graph $\mathcal{G}^*$ by conditioning the observed factors (using (Eq. 8) and (Eq. 17)).
 4:     Convert variables and factors to DLR canonical form. {Section 3.1}
 5:     Propagate DLR messages on $\mathcal{G}^*$. {Section H.2/Algorithm 3}
 6:     Conform the ancestral nodes to the belief: update the ensemble sample via a transformation $T$ so that $T(X_{\mathscr{Q}}^*) \sim b_{\mathcal{G}^*}(\mathbf{x}_{\mathscr{A}})$. {Section 3.3}
 7: **end while**
 8: **return** Approximate posterior sample $X_{\mathscr{Q}}^*$.

---

## H.2  GENBP FACTOR-TO-VARIABLE MESSAGE

We now outline the steps for a GEnBP $f_j \rightarrow \mathbf{x}_\ell$ factor-to-variable message when there is a single incoming message $\mathbf{x}_k \rightarrow f_j$. The outgoing message is Gaussian with canonical parameters $(\boldsymbol{n}'_\ell, P'_\ell)$, where $P'_\ell = U'_\ell - R'_\ell R'^\top_\ell$ is in DLR form. Extending the algorithm to multiple incoming messages is straightforward but verbose.

---

**Algorithm 3:** GEnBP $f_j \rightarrow \mathbf{x}_\ell$ Message (Single Incoming)

---

**Require:** Factor $f_j$ with canonical parameters:

- Information vector: $\boldsymbol{n} = \begin{bmatrix} \boldsymbol{n}_\ell \\ \boldsymbol{n}_k \end{bmatrix}$
- Precision matrix in DLR form: $P = U - RR^\top$, where

$$- \quad \mathrm{U} = \mathrm{diag}\left(\begin{bmatrix} \boldsymbol{u}_\ell \\ \boldsymbol{u}_k \end{bmatrix}\right),$$

$$- \quad \mathrm{R} = \begin{bmatrix} \mathrm{R}_\ell \\ \mathrm{R}_k \end{bmatrix} \text{ with } \mathrm{R}_\ell \in \mathbb{R}^{D_\ell \times N} \text{ and } \mathrm{R}_k \in \mathbb{R}^{D_k \times N}.$$

**Require:** Incoming message $\mathbf{x}_k \to f_j$ with canonical parameters:

- Information vector: $\boldsymbol{n}'_k \in \mathbb{R}^{D_k}$,
- Precision matrix in DLR form: $\mathrm{P}'_k = \mathrm{U}'_k - \mathrm{R}'_k \mathrm{R}'_k{}^\top$, where $\mathrm{U}'_k = \mathrm{diag}(\boldsymbol{u}'_k)$ and $\mathrm{R}'_k \in \mathbb{R}^{D_k \times N'}$.

**Ensure:** Outgoing message $f_j \to \mathbf{x}_\ell$ with canonical parameters:

- Information vector: $\boldsymbol{n}'_\ell$,
- Precision matrix in DLR form: $\mathrm{P}'_\ell = \mathrm{U}'_\ell - \mathrm{R}'_\ell \mathrm{R}'_\ell{}^\top$.

1: Combine the information vectors for $\mathbf{x}_k$: $\tilde{\boldsymbol{n}}_k \leftarrow \boldsymbol{n}_k + \boldsymbol{n}'_k$.
2: Combine the precision diagonals: $\tilde{\boldsymbol{u}}_k \leftarrow \boldsymbol{u}_k + \boldsymbol{u}'_k$.
3: Concatenate the low-rank components: $\tilde{\mathrm{R}}_k \leftarrow [\mathrm{R}_k \quad \mathrm{R}'_k]$.
4: Form the joint information vector: $\boldsymbol{n}_\Pi \leftarrow \begin{bmatrix} \boldsymbol{n}_\ell \\ \tilde{\boldsymbol{n}}_k \end{bmatrix}$.
5: Form the joint precision diagonal: $\mathrm{U}_\Pi \leftarrow \mathrm{diag}\left(\begin{bmatrix} \boldsymbol{u}_\ell \\ \tilde{\boldsymbol{u}}_k \end{bmatrix}\right)$.
6: Form the joint low-rank component: $\mathrm{R}_\Pi \leftarrow \begin{bmatrix} \mathrm{R}_\ell & 0 \\ \mathrm{R}_k & \mathrm{R}'_k \end{bmatrix}$.
7: Store DLR form joint precision matrix $\mathrm{P}_\Pi \leftarrow \mathrm{U}_\Pi - \mathrm{R}_\Pi \mathrm{R}_\Pi^\top$.
8: Invert to obtain the DLR joint covariance: $\mathrm{K}_\Pi \leftarrow \mathrm{P}_\Pi^{-1} = \mathrm{V}_\Pi + \mathrm{L}_\Pi \mathrm{L}_\Pi^\top$.
9: Compute the joint mean: $\boldsymbol{m}_\Pi \leftarrow \mathrm{K}_\Pi \boldsymbol{n}_\Pi = \mathrm{V}_\Pi \boldsymbol{n}_\Pi + \mathrm{L}_\Pi \mathrm{L}_\Pi^\top \boldsymbol{n}_\Pi$.
10: Extract the marginal mean for $\mathbf{x}_\ell$: $\boldsymbol{m}_\ell \leftarrow \boldsymbol{m}_\Pi[1{:}D_\ell]$.
11: Extract the marginal covariance for $\mathbf{x}_\ell$:

- Diagonal: $\mathrm{V}_\ell \leftarrow \mathrm{V}_\Pi[1{:}D_\ell, 1{:}D_\ell]$.
- Low-rank: $\mathrm{L}_\ell \leftarrow \mathrm{L}_\Pi[1{:}D_\ell, :]$.
- Store DLR-form $\mathrm{K}_\ell \leftarrow \mathrm{V}_\ell + \mathrm{L}_\ell \mathrm{L}_\ell^\top$.

12: Reduce the rank of $\mathrm{K}_\ell$ to $N$ via SVD:

- Compute SVD: $\mathrm{L}_\ell = \mathrm{ASB}^\top$.
- Retain the top $N$ components: $\mathrm{L}'_\ell = \mathrm{A}_{[:,1:N]} \mathrm{S}_{[1:N,1:N]}$.
- Store DLR covariance: $\mathrm{K}_\ell \leftarrow \mathrm{V}_\ell + \mathrm{L}'_\ell \mathrm{L}'_\ell{}^\top$.

13: Compute the DLR marginal precision: $\mathrm{P}'_\ell \leftarrow \mathrm{K}_\ell^{-1} = \mathrm{U}'_\ell - \mathrm{R}'_\ell \mathrm{R}'_\ell{}^\top$.
14: Compute the marginal information vector: $\boldsymbol{n}'_\ell \leftarrow \mathrm{P}'_\ell \boldsymbol{m}_\ell = \mathrm{U}'_\ell \boldsymbol{m}_\ell - \mathrm{R}'_\ell \mathrm{R}'_\ell{}^\top \boldsymbol{m}_\ell$.
15: **return** $\boldsymbol{n}'_\ell$ and $\mathrm{P}'_\ell = \mathrm{U}'_\ell - \mathrm{R}'_\ell \mathrm{R}'_\ell{}^\top$.

## H.3 ENSEMBLE RECOVERY

After belief propagation, suppose the belief for a query variable is $b(\mathbf{x}_\mathcal{Q}) \sim \mathcal{N}_M(\boldsymbol{x}_\mathcal{Q}; \boldsymbol{m}, \mathrm{K}_\mathcal{Q})$ with DLR covariance $\mathrm{K}_\mathcal{Q} = \mathrm{V}_\mathcal{Q} + \mathrm{L}_\mathcal{Q} \mathrm{L}_\mathcal{Q}^\top$. To convert this belief into ensemble samples, we choose an affine transformation $T$ that maps the prior ensemble $\mathrm{X}_\mathcal{Q}$ to an updated ensemble $\mathrm{X}'_\mathcal{Q} = T(\mathrm{X}_\mathcal{Q})$ so that the empirical distribution of the transformed ensemble approximates $\mathcal{N}_M(\boldsymbol{x}_\mathcal{Q}; \boldsymbol{m}, \mathrm{K}_\mathcal{Q})$ under a metric $d$. (For brevity we suppress the subscript $Q$.)

We restrict $T$ to the family of affine transformations

$$T_{\boldsymbol{\mu}, \mathrm{T}} : \mathrm{X} \mapsto \boldsymbol{\mu} \mathrm{B} + \check{\mathrm{X}} \mathrm{T},$$

with parameters $\boldsymbol{\mu} \in \mathbb{R}^D$ and $\mathrm{T} \in \mathbb{R}^{N \times N}$. Minimizing the distance between the empirical moments and the target moments yields

$$\boldsymbol{\mu} = \widehat{\mathbb{E}}\mathrm{X}, \tag{62}$$

$$\mathrm{T} = \arg\min_{\mathrm{T}} \left\| \widehat{\mathrm{Var}}_{\sigma^2}(\check{\mathrm{X}}\mathrm{T}) - \mathrm{K} \right\|_F^2 \tag{63}$$

$$= \arg\min_{\mathrm{T}} \left\| \widehat{\mathrm{Var}}_{\sigma^2}\big(T(\mathrm{X})\big) - \big(\mathrm{L}\mathrm{L}^\top + \mathrm{V}\big) \right\|_F^2. \tag{64}$$

Since the cost depends only on $\mathrm{T}\mathrm{T}^\top$, the minimizers are nonunique. We therefore optimize over $\mathrm{M} = \mathrm{T}\mathrm{T}^\top$ in the space of $N \times N$ positive semidefinite matrices $\mathbb{M}_+^N$:

$$\mathrm{M}^* = \arg\min_{\mathrm{M} \in \mathbb{M}_+^N} L(\mathrm{M}), \tag{65}$$

where

$$L(\mathrm{M}) := \left\| \frac{1}{N-1}\check{\mathrm{X}}\mathrm{M}\check{\mathrm{X}}^\top - \big(\mathrm{L}\mathrm{L}^\top + \tilde{\mathrm{V}}\big) \right\|_F^2, \tag{66}$$

with $\tilde{\mathrm{V}} := \mathrm{V} - \sigma^2 \mathrm{I}$. The gradient is given by

$$\nabla_{\mathrm{M}} L = \frac{2}{N-1}\check{\mathrm{X}}^\top \left( \frac{\check{\mathrm{X}}\mathrm{M}\check{\mathrm{X}}^\top}{N-1} - \mathrm{L}\mathrm{L}^\top - \tilde{\mathrm{V}} \right)\check{\mathrm{X}}.$$

An unconstrained solution $\mathrm{M}'$ satisfies

$$\check{\mathrm{X}}^\top \check{\mathrm{X}}\, \mathrm{M}'\, \check{\mathrm{X}}^\top \check{\mathrm{X}} = (N-1)\Big( \check{\mathrm{X}}^\top \mathrm{L}(\check{\mathrm{X}}^\top \mathrm{L})^\top + \check{\mathrm{X}}^\top \tilde{\mathrm{V}}\check{\mathrm{X}} \Big).$$

This system can be solved at $\mathcal{O}(N^3)$ cost (using, e.g., pivoted LDL decomposition) and then projected onto the PSD cone. Finally, we extract $\mathrm{T}$ via the SVD of $\mathrm{M}^* = \mathrm{U}\mathrm{S}\mathrm{U}^\top$ by setting $\mathrm{T} = \mathrm{U}(\mathrm{S}^+)^{1/2}$. With careful ordering, the total cost is $\mathcal{O}(N^3 + DN^2 + DM^2)$.

## I   ALTERNATIVE LINEARISATIONS AND LOW-RANK DECOMPOSITIONS

In response to a reader's question, we investigate whether GEnBP is merely a low-rank decomposition of the GaBP algorithm. We argue that it is not; rather, the relationship between the two is as illustrated in Figure 13.

Although both GEnBP and GaBP use Gaussian approximations, they do not employ the same approximations. This is why, as shown in Section 4, GEnBP can surpass GaBP in accuracy as well as speed. We noted in Appendix C.3 that GaBP not only selects a particular Gaussian density family (which it shares with GEnBP) but also uses specifically the $\delta$-method for approximating nonlinear factors, as opposed to the sample approximation of GEnBP. This apparently minor difference has significant implications, which we discuss below.

Consider the joint density of a factor $\mathbf{x}_2 = \mathcal{P}(\mathbf{x}_1)$ to highlight these differences. For simplicity, suppose that the input and output dimensions are equal, i.e. $D_1 = D_2 = D$. Two alternatives for covariance estimation have been discussed in this work.

GaBP uses propagation of errors:

$$\mathrm{Var}\begin{bmatrix} \mathbf{x}_1 \\ \mathbf{x}_2 \end{bmatrix} \simeq \begin{bmatrix} \mathrm{Var}\,\mathbf{x}_1 & J(\boldsymbol{m}_1)\,\mathrm{Var}\,\mathbf{x}_1 \\ \mathrm{Var}\,\mathbf{x}_1\,J(\boldsymbol{m}_1)^\top & J(\boldsymbol{m}_1)\,\mathrm{Var}\,\mathbf{x}_1\,J^\top(\boldsymbol{m}_1) \end{bmatrix} \tag{67}$$

where $J(\boldsymbol{m}_1)$ denotes the Jacobian of $\mathcal{P}$ evaluated at $\boldsymbol{m}_1$.

In contrast, GEnBP uses an empirical estimate:

$$\mathrm{Var}\begin{bmatrix} \mathbf{x}_1 \\ \mathbf{x}_2 \end{bmatrix} \simeq \widehat{\mathrm{Var}}_{\sigma^2\mathrm{I}} \begin{bmatrix} x^{(1)} & \cdots & x^{(N)} \\ \mathcal{P}(x^{(1)}) & \cdots & \mathcal{P}(x^{(N)}) \end{bmatrix}. \tag{68}$$

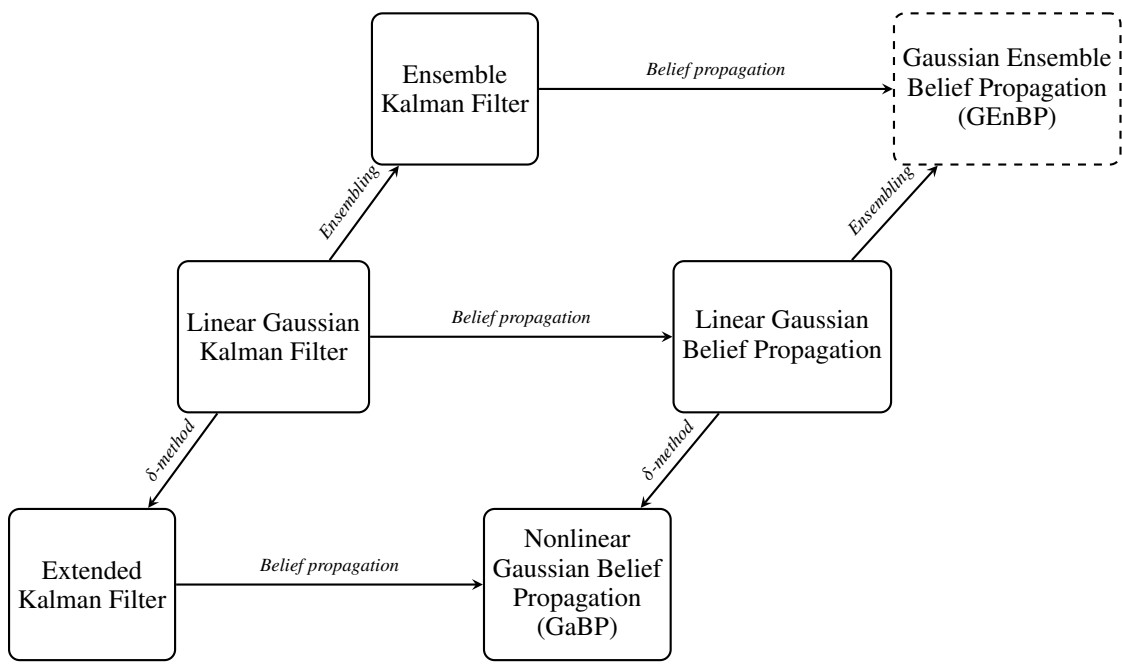

Figure 13: Relationship between GEnBP and GaBP.

Here the empirical covariance $\widehat{\mathrm{Var}}_{\sigma^2 \mathrm{I}}$ is computed using $N$ samples from the prior, with an inflation term $\sigma^2 \mathrm{I}$ to ensure invertibility.

As noted in Section 3.4, the cost of computing the empirical covariance in Equation 68 is higher than that of the propagation-of-errors estimate in Equation 67—both in time and space. There are two main components to this cost:

**Firstly**, the cost of generating the ensemble (in GEnBP) versus computing the Jacobians (in GaBP). In GEnBP, generating $N$ samples from the prior incurs a cost that scales as $\mathcal{O}(N)$. In contrast, computing the Jacobian in GaBP depends on the function but is generally $\mathcal{O}(D)$.

**Secondly**, the cost of performing the necessary matrix multiplications. The matrix product in Equation 67 involves three $D \times D$ matrices and hence costs $\mathcal{O}(D^3)$. In Equation 68, the empirical covariance is computed at a cost of $\mathcal{O}(DN)$. (It would be $\mathcal{O}(D^2 N)$ if we were to calculate the full covariance matrix; however, as argued in this paper, it suffices to compute the deviance matrices.)

For simplicity, suppose we are in a regime where $N \ll D$ and the node degree $K$ is small. In this setting, the belief propagation steps in GEnBP are generally cheaper. One might then ask whether GaBP could also benefit from low-rank belief updates in the SAEM setting.

Suppose we wished to construct a third option—a *low-rank GaPB* (LRGaBP)—which uses a low-rank decomposition of the covariance matrix to perform approximate GaBP in the hope of achieving efficiency similar to GEnBP. First, one would compute a rank $N$ decomposition of the prior covariance, i.e.

$$\mathrm{Var}\,\mathbf{x}_1 \approx \mathrm{LL}^\top, \quad \mathrm{L} \in \mathbb{R}^{D \times N},$$

perhaps by eigendecomposition. Naively, this operation is $D^3$, or $\mathcal{O}(D^2 \log N + N^2 D + N^3)$ using the method of (Halko et al., 2010, 6.1).

The joint variance arising from the propagation-of-errors (or $\delta$-method (Dorfman, 1938)) can then be written as

$$\mathrm{Var}\begin{bmatrix}\mathbf{x}_1\\\mathbf{x}_2\end{bmatrix} \simeq \begin{bmatrix} \mathrm{Var}\,\mathbf{x}_1 & J(\boldsymbol{m}_1)\,\mathrm{Var}\,\mathbf{x}_1 \\ \mathrm{Var}\,\mathbf{x}_1\,J(\boldsymbol{m}_1)^\top & J(\boldsymbol{m}_1)\,\mathrm{Var}\,\mathbf{x}_1\,J^\top(\boldsymbol{m}_1) \end{bmatrix}$$

$$= \begin{bmatrix} \mathrm{LL}^\top & J(\boldsymbol{m}_1)\,\mathrm{LL}^\top \\ \mathrm{LL}^\top\,J(\boldsymbol{m}_1)^\top & J(\boldsymbol{m}_1)\,\mathrm{LL}^\top\,J^\top(\boldsymbol{m}_1) \end{bmatrix}$$

$$= \begin{bmatrix} \mathrm{L} \\ J(\boldsymbol{m}_1)\,\mathrm{L} \end{bmatrix} \begin{bmatrix} \mathrm{L} \\ J(\boldsymbol{m}_1)\,\mathrm{L} \end{bmatrix}^\top. \tag{69}$$

This is indeed a low-rank decomposition and is amenable to the same techniques as the other low-rank methods outlined in Appendix F. However, it is not computationally competitive. Since $J(\boldsymbol{m}_1)$ is a $D \times D$ matrix, the product $J(\boldsymbol{m}_1)\,\mathrm{L}$ costs $\mathcal{O}(D^2 N)$, in addition to the $\mathcal{O}(D)$ cost of computing the Jacobian.

We summarise the costs of this hypothetical LRGaBP step in Table 3.

Notably, although the precise relationship between these algorithms depends on the problem structure, we generally expect GEnBP to be more efficient than LRGaBP for high-dimensional problems at a fixed $N$; in many cases GEnBP exhibits a lower exponent in $D$. Furthermore, since LRGaBP is an approximation to GaBP, its accuracy is inherently bounded by that of GaBP, whereas GEnBP—being a different approximation to the target estimand—is not subject to this limitation.

Nonetheless, the LRGaBP algorithm has not been pursued further given its computational inefficiency and limited accuracy gains relative to GEnBP.

Table 3: Computational Costs for Gaussian Belief Propagation (GaBP), Ensemble Belief Propagation (GEnBP), and the hypothetical Low-Rank Gaussian Belief Propagation (LRGaBP) for node dimension $D$ and ensemble size/component rank $N$.

| Operation | GaBP | GEnBP | LRGaBP |
|---|---|---|---|
| **Time** | | | |
| Simulation | $\mathcal{O}(1)$ | $\mathcal{O}(N)$ | $\mathcal{O}(1)$ |
| Error propagation | $\mathcal{O}(D^3)$ | — | $\mathcal{O}(D^2 N)$ |
| Jacobian calculation | $\mathcal{O}(D)$ | — | $\mathcal{O}(D)$ |
| Covariance SVD | — | $\mathcal{O}(N^3 + N^2 D)$ | $\mathcal{O}(D^2 \log N + N^3 + N^2 D)$ |
| **Space** | | | |
| Covariance Matrix | $\mathcal{O}(D^2)$ | $\mathcal{O}(ND)$ | $\mathcal{O}(ND + D^3)$ |
| Precision Matrix | $\mathcal{O}(D^2)$ | $\mathcal{O}(ND)$ | $\mathcal{O}(ND)$ |

