# OpenReview forum: "Gaussian Ensemble Belief Propagation for Efficient Inference in High-Dimensional, Black-box Systems"
_ICLR.cc/2025/Conference — ICLR 2025 Poster_

### Official Review · Reviewer_QpY9 · 2024-10-28

**Soundness:** 3
**Presentation:** 2
**Contribution:** 4
**Rating:** 6
**Confidence:** 3

**Summary:**

Efficient inference is challenging in graphical models with high-dimensional variables.
The authors introduce an inference algorithm based on Gaussian Ensemble Belief Propagation (GEnBP) and the Ensemble Kalman Filter (EnKF), combining strengths of both algorithms to arrive at an algorithm which is competitive in accuracy, in scalability with respect to the dimensionality of the variables, and which can be applied to a wide range of inference problem structures.

**Strengths:**

The authors present a novel methodology with well motivated approximations and algorithmic structure, where important connections and details are highlighted. The methodology contains contributions and explanations that are valuable also on their own. Graphical illustrations and notation is used in an effective manner to improve clarity, and the text is well structured.

**Weaknesses:**

Although I find the text to be well structured in terms of its content, the text contains many language errors, forgotten punctuation, and incorrect figure references. I have listed some errors I encountered below which the authors may find helpful for improvement of the manuscript.

List errors/suggestions style and language:
- Line 22: no spaces around “–”.
- Line 48: why “seems” novel and not “is” novel, if that is the case?
- Line 68: “table” is not capitalised.  This is also the case in many other places. Note that it should be capitalised since “Table 1” is a name.
- Line 71: “section” not capitalised.  This is also the case in many other places.
- Line 72: missing punctuation after “approximations”.
- Line 81-95: In my opinion, excessive use of bolds. Is it needed more than say a couple of times here, or even at all?
- Line 83: How about “a novel message-passing method for inference in high-dimensional graphical models”?
- Line 91: “Rank” is (incorrectly) capitalised even though the list is written as a sentence.
- Line 93: Same as above but for “Compute”.
- Line 99: How about “We will now introduce our notation and essential concepts”?
- Line 100: “appendix” is not capitalized. This is also the case in many other places.
- Line 118: How about “We will assume that queries are always ancestral variables, i.e. that …”?
- Line 134: “equation” is not capitalized.  This is also the case in many other places.
- Line 146: Missing word 'with'. “Edges connect each factor node j with each variable node…”?
- Line 173: Unnecessary abbreviations, “we’ve” and “we’re”.
- Line 180: observed is misspelt.
- Line 184: “figure” not capitalised.  This is also the case in many other places.
- Line 191: Missing punctuation before “The following”.
- Line 202: “definition” not capitalised. This is also the case in many other places.
- Line 227: Double punctuation.
- Line 261: Missing punctuation before “Marginalization”.
- Line 294: Missing punctuation before “Throughout”.
- Line 312: Missing punctuation before “Here”.
- Line 314: Misspelt equivalently.
- Line 323: Incorrect grammar, “See appendix I for a comparison with a naive attempt to do without the ensemble”.
- Line 325: Missing punctuation before “Then”.
- Line 336: Missing punctuation after “high dimensions”.
- Line 353: “algorithm” not capitalised. This is also the case in many other places.
- Line 376: Incorrect grammar: “For a variable with K neighbors, in the worst case M = KN , when the cost becomes…” and missing punctuation.
- Line 381: How about “In many practical applications it holds that N ≪ D, resulting in significant computational savings compared to GaBP which scales poorly with D.”.
- Line 385: How about “better suited”?
- Line 404: Incorrect grammar “While GEnBP scales favorably with D, but unfavourably with respect to the node degree K, scaling as O(K 3 ) for some operations.”


The presentation of the experiments is unclear at times, please see Questions.

**Questions:**

Regarding Section 4.1:
- Figure 2 is never referenced. Although I assume the first paragraph describes observations made from it?
- Why is the log likelihood only shown for GaBP in the case of the lowest dimensionality setting used?
- I struggle to reach the following interpretation based on Figure 2: “We see that the Laplace approximation … but its posterior likelihood, while similar to GaBP, is even less stable, and both are inferior to GEnBP”:
- - For dimensionalities greater than around 100, it seems GEnBP is underperforming Laplace in terms of log likelihood (while the results provides no comparison to GaBP)?
- - Why is GaBP and Laplace interpreted as less stable than GEnBP based on this figure?

Section 4.2:
- Figure 3 is from the text implied to contain results using Langevin MC, but it does not --- only Figure 4 does.
- The results show (and the text says) that GaBP performs better in terms of log likelihood for high viscosity settings. What do you suspect is the explanation for this?

Overall, I think the paper is good and makes valuable contributions. However, I think the manuscript needs an overhaul to fix the language errors, missing punctuation, and figure references before it is ready for publication. If this is done and the experiments section is clarified I intend to raise my evaluation score.

---

> ### Author Response · Authors · 2024-11-15
> **Part 1a: Style and language**
>
> Thanks very much for your detailed review. We are glad that you think the paper makes valuable contributions, and presents a novel methodology with well motivated approximations and algorithmic structure.
> We address your individual comments below.
> We hope that this satisfies your concerns about clarity of the experiments and language errors.
>
> - > Line 22: no spaces around “–”.
>     - Fixed, and now correctly use an em dash ---
> - Line 48: why “seems” novel and not “is” novel, if that is the case?
>     - Indeed, their combination *is* novel, as demonstrated in the manuscript and mentioned by you and other reviewers. We have corrected this.
> - Line 68: “table” is not capitalised. This is also the case in many other places. Note that it should be capitalised since “Table 1” is a name.
>     - We have updated the manuscript so that Table, Section, Figure, Appendix, Equation, Definition, and Algorithm are all upper-case, as they are names.
> - Line 71: “section” not capitalised. This is also the case in many other places.
>     - We have updated the manuscript so that Table, Section, Figure, Appendix, Equation, Definition, and Algorithm are all upper-case, as they are names.
> - Line 72: missing punctuation after “approximations”.
>     - We have added a full stop after "approximations".
> - Line 81-95: In my opinion, excessive use of bolds. Is it needed more than say a couple of times here, or even at all?
>     - Agreed. We have removed the bolds, as the itemised list already makes the statements stand out sufficiently well.
> - Line 83: How about “a novel message-passing method for inference in high-dimensional graphical models”?
>     - Agree this wording is cleaner. Have updated this phrase as you suggest.
> - Line 91: “Rank” is (incorrectly) capitalised even though the list is written as a sentence.
>     -We have fixed capitalisation of "rank"
> - Line 93: Same as above but for “Compute”.
>     - We have fixed capitalisation of "compute"
> - Line 99: How about “We will now introduce our notation and essential concepts”?
>     - Fixed, as per your suggestion.
> - Line 100: “appendix” is not capitalized. This is also the case in many other places.
>     - We have updated the manuscript so that Table, Section, Figure, Appendix, Equation, Definition, and Algorithm are all upper-case, as they are names.
> - Line 118: How about “We will assume that queries are always ancestral variables, i.e. that …”?
>     - Agree that this better describes our setting. We have updated as you suggest.
> - Line 134: “equation” is not capitalized. This is also the case in many other places.
>     - We have updated the manuscript so that Table, Section, Figure, Appendix, Equation, Definition, and Algorithm are all upper-case, as they are names.
> - Line 146: Missing word 'with'. “Edges connect each factor node j with each variable node…”?
>     - Well-spotted. We have added the missing "with".
> - Line 173: Unnecessary abbreviations, “we’ve” and “we’re”.
>     - Removed abbreviations and contractions so "we've" is now we have and "we're" is now "we are".
> - Line 180: observed is misspelt.
>     - Updated to correct spell "observed".
> - Line 184: “figure” not capitalised. This is also the case in many other places.
>     - We have updated the manuscript so that Table, Section, Figure, Appendix, Equation, Definition, and Algorithm are all upper-case, as they are names.
> - Line 191: Missing punctuation before “The following”.
>     - Added missing full stop before "The following".
> - Line 202: “definition” not capitalised. This is also the case in many other places.
>     - We have updated the manuscript so that Table, Section, Figure, Appendix, Equation, Definition, and Algorithm are all upper-case, as they are names.
> - Line 227: Double punctuation.
>     - Removed the extra full stop.
> - Line 261: Missing punctuation before “Marginalization”.
>     - Added the missing full stop before "Marginalization".
> - Line 294: Missing punctuation before “Throughout”.
>     - Verified that in the updated manuscript, "Throughout" is preceeded by a full stop.
> - Line 312: Missing punctuation before “Here”.
>     - Verified correct placement of punctuation before "Here".
> - Line 314: Misspelt equivalently.
>     - Fixed spelling of equivalently.
> - Line 323: Incorrect grammar, “See appendix I for a comparison with a naive attempt to do without the ensemble”.
>     - Whilst there is a grammatical parse of this sentence, we agree that it is unnecessarily difficult to deduce it. Updated to read “Appendix 1 compares the computational cost of the GEnBP with an attempt to exploit DLR factorisations without using an ensemble.”
> - Line 325: Missing punctuation before “Then”.
>     - Added full stop before "Then".
> - Line 336: Missing punctuation after “high dimensions”.
>     - Added full stop after "high dimensions".

---

> > ### Author Response · Authors · 2024-11-15
> > **Part 1b: Style and language**
> >
> > Continuing part 1a
> >
> > - Line 353: “algorithm” not capitalised. This is also the case in many other places.
> >     - We have updated the manuscript so that Table, Section, Figure, Appendix, Equation, Definition, and Algorithm are all upper-case, as they are names.
> > - Line 376: Incorrect grammar: “For a variable with K neighbors, in the worst case M = KN , when the cost becomes…” and missing punctuation.
> >     - Corrected this sentence to read "For a variable with $K$ neighbors, in the worst case $M = K N$, and the cost becomes $\mathcal{O}(N^3 + D N^2 K^2)$."
> > - Line 381: How about “In many practical applications it holds that N ≪ D, resulting in significant computational savings compared to GaBP which scales poorly with D.”.
> >     - Changed as per your suggestion.
> > - Line 385: How about “better suited”?
> >     - Changed as per your suggestion.
> > - Line 404/405: "While GEnBP scales favorably with D, but unfavourably with respect to the node degree K, scaling as O(K3) for some operation
> >     - Agreed, this is unnecessarily muddy. We have update it to read “GEnBP operations scale favourably with $D$, at worst $\mathcal{O}(D)$, compared to the worst $\mathcal{O}(D^3)$ for GaBP. They scale unfavourably with respect to the node degree, at worst $\mathcal{O}(K^3)$, compared to GaBP’s $\mathcal{O}(1)$.”
> >
> > We look forward to further engaging with you during the rebuttal period, should you have any additional queries. Please let us know if you have any further concerns.

---

> > > ### Comment · Reviewer_QpY9 · 2024-11-16
> > >
> > > Thank you for resolving the listed language errors.
> > >
> > > I would appreciate answers to my questions in the "Questions" section, some of which are also raised by other reviewers.

---

> > > > ### Author Response · Authors · 2024-11-19
> > > > **Part 2: substntive matters**
> > > >
> > > > Naturally! we intend to answer all of the questions, and we appreciate the reviewer's timely engagement.
> > > > The next batch follows:
> > > >
> > > > >Regarding Section 4.1:
> > > > >
> > > > > Figure 2 is never referenced. Although I assume the first paragraph describes observations made from it?
> > > >
> > > > Good catch, thank you. We have added the line "Figure 2 plots the influence of dimension upon various measures of inference quality." to section 4.1.
> > > >
> > > > >  Why is the log likelihood only shown for GaBP in the case of the lowest dimensionality setting used?
> > > >
> > > > Good question. Another reviewer has asked this also.
> > > > In answer: There are two reasons. Firstly, there seem to be fewer log likelihood error bars than there are due to overlapping values where the estimates are very similar. We have addressed this by offsetting the plots for different values. The second is that the log-likelihood is simply undefined for many values of D, in that the numerically-calculated Hessian in the case of Laplace, or posterior covariance in the case of GaBP, fails to be positive-definite. We have added a clarification to the caption: "Missing log likelihood values denote undefined values arising from non-positive-definite covariance estimates."
> > > >
> > > > > I struggle to reach the following interpretation based on Figure 2: “We see that the Laplace approximation … but its posterior likelihood, while similar to GaBP, is even less stable, and both are inferior to GEnBP”:
> > > > >        For dimensionalities greater than around 100, it seems GEnBP is underperforming Laplace in terms of log likelihood (while the results provides no comparison to GaBP)?
> > > > >>        Why is GaBP and Laplace interpreted as less stable than GEnBP based on this figure?
> > > >
> > > > A reasonable question; our phrasing there contains an error, and the figure is poorly labeled. We apologies for the confusing and thank our reviewer for catching the mistake.
> > > >
> > > > We hope that the previous answer has clarified this somewhat, but we think it is wise to expand at greater length:
> > > > We propose that it would be reasonable to evaluate the posterior log likelihood for all methods at the ground truth by two qualities
> > > > 1. that it is high, and
> > > > 2. that is is actually defined.
> > > > One of the strengths of the Ensemble Kalman Filter is that  the posterior covariance matrix is nearly always positive definite, and so the log likelihood is nearly always defined; the gradient based methods frequently produce non-positive definite covariance estimates and hence undefined log-likelihoods; it is this latter property that we were referring to as “stability” although since we did not define that term, it is understandable the reviewer had questions about it.
> > > >
> > > > We have amended the manuscript to clarify this point with the following two revisions
> > > >
> > > > 1. altering the text to be both clearer and more correct with the following rewrite:
> > > >
> > > >    >At small dimension both Laplace and GEnBP produce similar log likelihood estimates, with GaBP somewhat worse than either. As the dimension increases beyond 100, however, the gradient-based covariances in both GaBP and Laplace methods are increasingly likely not to be positive-definite, and thus the posterior log-likelihood may fail to be defined. GEnBP by contrast decays more slowly in log likelihood, and is always defined.
> > > >
> > > > 2. amending the plot: although the GaBP does have many missing log likelihood values, we should see more than only one; this is a glitch. We are preparing a revised version of the figure.
> > > >
> > > > Do these changes address the reviewer's concerns?
> > > >
> > > > >Section 4.2:
> > > > >
> > > > >    Figure 3 is from the text implied to contain results using Langevin MC, but it does not --- only Figure 4 does.
> > > >
> > > > Thank you for picking up this  error.
> > > > We missed the Langevin data from the submitted version of the paper.
> > > > As can be inferred from the text the figure we attempted to include shows a generally good result from the Langevin sampler but at high computational cost.
> > > > The revised PDF fixes this mistake.
> > > >
> > > > >    The results show (and the text says) that GaBP performs better in terms of log likelihood for high viscosity settings. What do you suspect is the explanation for this?
> > > >
> > > > There are two questions here, if we understand correctly.
> > > >
> > > > 1. Why does the viscosity parameter affect the log likelihood the way that it does for each inference method?
> > > > 2. is there something suspicious about the GaBP result achieving good log likelihood even while it achieves a poor RMSE?
> > > >
> > > > In our opinion, question #2 suggests something suspicious about the GaBP result; we shouldn't see the RMSE and the log likelihood diverge so much for the GaBP result.
> > > > We are re-running the experiments to sanity check this , and will return to both these quwestions as soon as the run are complete; this will take a day or two. We beg the reviewer's patience on this final point.

---

> > > > > ### Comment · Reviewer_QpY9 · 2024-11-25
> > > > >
> > > > > Thank you. I have increased my score.

---

> ### Author Response · Authors · 2024-11-22
> **We worked out what was suspicious about the viscosity graph**
>
> Thanks for drawing our attention to the right hand side of the viscosity graph! We have resolved #2 by uncovering a problem with our experiment: specifically, for viscosity values of approximate 10 and higher, none of the methods successfully recover a value for the mean. Instead, all converge to various spurious optima with wide posterior variance; the results in the $10\le\eta\leq10^2$ region of the graph are all essentially equally meaningless as a test of the inference.
> It seems that in this domain, the problem is not well posed, in the sense  that there is not much information in the slow-moving simulation data for the inference methods to resolve. We are inclined to simply delete the region of the graph where the problem is intractable with $\eta\ge 10$, since it is not informative about the primary question of the paper, the method comparison. A more elegant solution would be sweep the system behaviour in terms of the Reynolds number rather than viscosity, which will reduce the confounding in the system. We leave that aside for the moment, as it is not the central concern of the paper. We thank the reviewer for identifying this issue, which has improved the quality of our results.

---

### Official Review · Reviewer_Mt1V · 2024-10-31

**Soundness:** 4
**Presentation:** 4
**Contribution:** 3
**Rating:** 8
**Confidence:** 4

**Summary:**

The paper presents a generalisation of the ensemble Kalman filter to make inference on arbitrary graphical models using belief propagation  (BP) with ensembles. By relying on ensembles rather than keeping track of the full moments, the method is able to scale to problems where the state dimension of the nodes are high dimensional, e.g. weather forecasting. The authors propose several numerical tricks that make this possible while retaining scalability. This is tested against a standard Gaussian BP on some examples, showcasing its computational efficiency and performance improvements, offered by its improved handling of nonlinear conditional dependence between nodes.

**Strengths:**

The paper is generally well-written and well-motivated. The methodology presented, while seemingly a small step beyond standard BP, is not at all trivial to execute, requiring various tricks to make it work in practice. In particular, I find the efficient computation of the factor-to-variable message and the ensemble conformation step to be an interesting contribution that is essential to retain the ensemble-based representation of the beliefs. The experiments showcase interesting settings where this technique could be applied, such as latent forcing identification in fluid models. In addition, it shows clear benefits of the approach over vanilla BP, with orders of magnitude increase in computation speed and performance gains.

**Weaknesses:**

While the paper is clearly written for the most part, there are some parts that I find need more explanation. See my questions below for details. I also find that the experiments can be improved with further baseline comparisons. Is GaBP really the only baseline that can solve the system identification problems in the experiments? One may probably also consider methods like the integrated nested Laplace approximation (INLA), or particle MC methods (e.g. the particle marginal Metropolis-Hastings algorithm).

**Questions:**

- It would be great if the authors could please clarify why we need a particle representation of the beliefs when we can just make computations with the DLR representation? My understanding is that in EnKF, it is used to propagate the particles by the nonlinear dynamics model and update them using Matheron's rule, but what does this correspond to in the general BP framework (in terms of steps (5)-(7))?
- Do we need tricks like covariance localisation and inflation in this general BP setting? These are almost always used to make EnKF work robustly in high dimensions.
- In Figure 2, why are some results missing for GaBP and Laplace method in the log-likelihood comparison? Was it producing NaNs due to overconfident predictions?

---

> ### Author Response · Authors · 2024-11-21
> **Weaknesses**
>
> > While the paper is clearly written for the most part, there are some parts that I find need more explanation. See my questions below for details. I also find that the experiments can be improved with further baseline comparisons. Is GaBP really the only baseline that can solve the system identification problems in the experiments? One may probably also consider methods like the integrated nested Laplace approximation (INLA), or particle MC methods (e.g. the particle marginal Metropolis-Hastings algorithm).
>
> We thank the reviewer for their insightful comments and suggestions, and the chance to clarify our position.
> Indeed, in the class of system identifications problems, GaBP is not the only method that can be used; there are many competitors in that domain. PMMH would indeed be worth exploring.
> This question has been raised by other reviewers, so we have revised the introduction and put some additional general comment in Common Response 1 on this theme.
> If we consider the class of belief propagation methods, i.e those which can generalise beyond the system identification setting, then there are few that handle the high-dimensional nodes that we consider here.
>
> The reviewer has raised an singular point by mentioning INLA, which uses a different approach to attain scalability.
> INLA does indeed handle high dimensional nodes with low-rank correlation induced by the SPDE dynamics, which leads to a non-trivial degree of overlap with GEnBP, and can also attain a high degree of efficiency by exploiting sparsity in precision matrix representations.
> It would be interesting to see if the posterior precision matrix generated by a Navier-Stokes dynamics could be approximated within the sparse precision matrix framework of INLA.
> Nonetheless, such an extension of INLA does not seem to us to be trivial.
> Related techniques such as fixed-rank kriging also show promise in this domain.
> We believe that within our setting the extension of INLA methods to high-dimensional nodes would be a non-trivial task, and so we have not pursued any experiments along these lines. However, this does merit mention in the introduction, and we have included a reference to this work.
>
> We will extend the introduction with mentions of methods to handle spatial random fields by Gauss-Markov random fields, and the use of fixed-rank kriging in the introduction.
>
> * Bakka, H., Rue, H., Fuglstad, G.-A., Riebler, A., Bolin, D., Illian, J., Krainski, E., Simpson, D., & Lindgren, F. (2018). Spatial modeling with R-INLA: A review. WIREs Computational Statistics, 10(6), e1443. https://doi.org/10.1002/wics.1443
> * Cressie, N., Shi, T., & Kang, E. L. (2010). Fixed Rank Filtering for Spatio-Temporal Data. Journal of Computational and Graphical Statistics, 19(3), 724–745.
> * Lindgren, F., & Rue, H. (2015). Bayesian Spatial Modelling with R-INLA. Journal of Statistical Software, 63(i19), 1–25. https://doi.org/10.18637/jss.v063.i19
> * Lindgren, F., Rue, H., & Lindström, J. (2011). An explicit link between Gaussian fields and Gaussian Markov random fields: The stochastic partial differential equation approach. Journal of the Royal Statistical Society: Series B (Statistical Methodology), 73(4), 423–498. https://doi.org/10.1111/j.1467-9868.2011.00777.x
> * Sigrist, F., Künsch, H. R., & Stahel, W. A. (2015). Stochastic partial differential equation based modelling of large space-time data sets. Journal of the Royal Statistical Society: Series B (Statistical Methodology), 77(1), 3–33. https://doi.org/10.1111/rssb.12061

---

> ### Author Response · Authors · 2024-11-21
> **Q1**
>
> ### Q1
>
> > It would be great if the authors could please clarify why we need a particle representation of the beliefs when we can just make computations with the DLR representation? My understanding is that in EnKF, it is used to propagate the particles by the nonlinear dynamics model and update them using Matheron's rule, but what does this correspond to in the general BP framework (in terms of steps (5)-(7))?
>
> We thank the reviewer for the chance to build intuition around this point.
> The particle-based representation analogously _plays  the same role as the linearization step does in traditional GaBP_.
> We suspect that as with traditional GaBP, various schedules of solving the least squares problem (by belief propagation) and quasi-relinearization (by sampling) might produce different computational and statistical trade-offs.
> We have avoided investigating that question here, as the paper is dense enough introducing the basic concepts, but in a full-length coverage of the method, this would be a an important question.
> Our reviewer is not the first person to ask about this; We direct your attention to Figure 11 in the Appendix, which makes the analogy diagrammatic.
>
> Further, one might wonder if one need the particles at all, or if one could just use the DLR representation directly. The answer to this turns out to be no, but this might not be immediately clear. A detailed explanation is given in appendix I, where we show that the convenient preservation of low-rank factorizations is in some sense special to the ensemble statistics. As such the two representations are inextricably coupled.
>
> We hope that addresses the reviewer's question? If not, we stand ready to address any further queries on this point.

---

> ### Author Response · Authors · 2024-11-21
> **Q2**
>
> > Do we need tricks like covariance localisation and inflation in this general BP setting? These are almost always used to make EnKF work robustly in high dimensions.
>
> A good pragmatic question, which we briefly investigated in an early implementation of the method.
> We suspect that the answer to this is a simple "yes", although we have not investigated it in any depth.
> More generally we suspect that the same tricks that are used in EnKF would be useful in GEnBP, and moreover many of the same tricks that are used in GaBP would be useful in GEnBP.
> Ensemble or DLR equivalents of covariance localisation, covariance inflation and adaptive ensemble selection from EnKF seem relatively simple to transfer to GEnBP, and we suspect that they would be useful in practice.
> Similarly, from GaBP, the use of message damping, incremental updating, robust covariance scaling and truncation of graphs would likely extend the range of applicability of the method by providing increased stability and convergence at lower cost if adequately tuned.
>
> That said, taken together, the number of such tricks is large, and we have not investigated such connections in this work as it seems that the primary goal of introducing the method is already dense for a conference paper.
> For now, we are happy to mention this in the conclusion and further work section; as such we have appended the previous paragraph to the paper's conclusion, replacing the existing sentence "Future work includes integrating improvements from GaBP and EnKF literature, such as graph pruning for computational efficiency and robust distributions to handle outliers"

---

> ### Author Response · Authors · 2024-11-21
> **Q3**
>
> ### Q3
>
> > In Figure 2, why are some results missing for GaBP and Laplace method in the log-likelihood comparison? Was it producing NaNs due to overconfident predictions?
>
> Exactly so, and we have modified the captions accordingly. However, in light of questions about this from multiple reviewers, we are preparing a new version of the figure; rather than letting one NaN poison the whole estimate, we will cull NaNs from the dataset, which gains us some additional data points.
> We will update the reviews shortly with "Common Response 2" which explain the changed methodology, and include the new figure in the revised manuscript.

---

> > ### Comment · Reviewer_Mt1V · 2024-11-22
> > **Thank you for the responses**
> >
> > We thank the authors for the detailed responses to my questions. Indeed, I have missed that INLA and PMMH only deals with learning low dimensional parameters - here you are dealing with a high dimensional case, so these techniques won't apply here. Thanks for also clarifying the importance of the particle representation. I will retain my original score of 8.

---

### Official Review · Reviewer_HdeM · 2024-11-02

**Soundness:** 3
**Presentation:** 3
**Contribution:** 3
**Rating:** 8
**Confidence:** 3

**Summary:**

The paper introduces the GEnBP algorithm, a method that combines Ensemble Kalman Filter (EnKF) and Gaussian Belief Propagation (GaBP) to efficiently perform inference in high-dimensional systems. GEnBP updates ensembles of prior samples into posterior samples using a message-passing strategy on a graphical model, making it suitable for applications with complex dependencies and high-dimensional variables, such as in geospatial and physical models.

**Strengths:**

The paper is well-structured, making it easy to follow the ideas presented. The proposed method can be considered to be novel in its approach to addressing the computational challenges of GaBP in high-dimensional systems.

**Weaknesses:**

While GEnBP demonstrates strong performance in high-dimensional contexts, its complexity scales disproportionately with the node degree in graphical models. This presents a significant limitation in highly connected graphs, where methods such as Forney factorization have been proposed by the authors as partial solutions. However, the effectiveness of these methods remains uncertain. Can some simple numerical results or insightful analyses be added to showcase this potential solution?

Additionally, a notable drawback is the absence of comparative analysis with other related works, both those employing and not employing EnKF, for efficient inference in high-dimensional, black-box systems. Key related works include:

  - Chen Y, Sanz-Alonso D, Willett R. "Autodifferentiable Ensemble Kalman Filters," *SIAM Journal on Mathematics of Data Science*, 2022; 4(2):801-833.

  - Chen Y, Sanz-Alonso D, Willett R. "Reduced-order Autodifferentiable Ensemble Kalman Filters," *Inverse Problems*, 2023; 39(12):124001.

  - Lin Z, Sun Y, Yin F, Thiéry A. "Ensemble Kalman Filtering-Aided Variational Inference for Gaussian Process State-Space Models." *arXiv preprint arXiv:2312.05910*, 2023.

  - Girin L, Leglaive S, Bie X, Diard J, Hueber T, Alameda-Pineda X. "Dynamical variational autoencoders: A comprehensive review." *arXiv preprint arXiv:2008.12595*, 2020.

Including a discussion or comparison on system identification and data assimilation from these (and other existing) works would enhance the comprehensiveness of the proposed method's positioning.

**Questions:**

1. Figure 2 is not cited in the text, which may confuse readers
2. Consider put Figure 1 in the experimental section
3. The paper highlights potential applications in fields like geospatial prediction and physical model inversion, which may involve real-time data. How does GEnBP handle streaming data, and what would be required to extend it to real-time applications with time-varying dependencies?

---

> ### Author Response · Authors · 2024-11-20
> **Weaknesses Part 1**
>
> > While GEnBP demonstrates strong performance in high-dimensional contexts, its complexity scales disproportionately with the node degree in graphical models. This presents a significant limitation in highly connected graphs, where methods such as Forney factorization have been proposed by the authors as partial solutions. However, the effectiveness of these methods remains uncertain. Can some simple numerical results or insightful analyses be added to showcase this potential solution?
>
>
> Dear Reviewer,
>
> Thank you for highlighting the challenges related to the applicability of our algorithm across different graphical structures. We acknowledge that characterizing suitable graphical structures for belief propagation methods remains a pervasive challenge in the field, particularly outside of exponential families where few analytic results exist.
>
> In recognition of these complexities, we have carefully avoided claiming that our method is universally applicable across all graph structures. As noted, the literature already extensively explores these themes, and we anticipate future work will continue to shed light on these issues.
>
> To ensure clarity and transparency about the scope and applicability of our method, we have provided evidence within the paper that demonstrates GEnBP's competitiveness in specific scenarios. We refer you to the note in our conclusion, which states:
>
> “Comparing the overall computational cost of the algorithms is complex and heavily dependent on the graph structure. Our empirical results demonstrate scenarios where GEnBP is much more efficient than GaBP, though we have not characterized the full range of problems where this advantage holds.”
>
> **Proposed Revision:** To better reflect the experimental nature of our findings, we propose modifying the abstract as follows:
>
> From:
> “GEnBP is particularly advantageous when the ensemble size is considerably smaller than the inference dimension.”
>
> To:
> “GEnBP has shown particular advantages when the ensemble size is considerably smaller than the inference dimension _in our experiments_.”
>
> We believe this amendment will more accurately represent the experimental conditions under which GEnBP's benefits were observed.
>
> We appreciate your insightful feedback and look forward to refining our manuscript accordingly.

---

> ### Author Response · Authors · 2024-11-21
> **Weaknesses Part 2**
>
> > Additionally, a notable drawback is the absence of comparative analysis with other related works, both those employing and not employing EnKF, for efficient inference in high-dimensional, black-box systems.
>
> We thank the reviewers for this opportunity to clarify the positioning of our work in the literature.
> See also our Common Response 1 for our general stance regarding system identification and state-filtering methods. In short, while we acknowledge there are many such, our goal is rater to construct a general belief-propagation method method which does not introduce the assumption that the model is of system identification type.
>
> On that basis we would expect the low-rank factorizations that we exploit here could be complementary to other methods with more restrictive assumptions. To visit the cited references:
>
> > Key related works include:
> >
> > * Chen Y, Sanz-Alonso D, Willett R. "Autodifferentiable Ensemble Kalman Filters," SIAM Journal on Mathematics of Data Science, 2022; 4(2):801-833.
> > * Chen Y, Sanz-Alonso D, Willett R. "Reduced-order Autodifferentiable Ensemble Kalman Filters," Inverse Problems, 2023; 39(12):124001.
>
> Thank you for both of these references.
> We concentrate on the latter of these two references, which is the most general and also the most relevant to our work.
> This paper has  commonalities with out work; in particular in section 3.2, they introduce a low-rank empirical factorization of the covariance matrix, which is a key component of our work.
> Their method in algorithm 3.1 still relies upon a state-space-type graphical structures as in their Figure 2, which is to say, they do no derive a factor-to-node message-passing algorithm as we do.
> On the other hand, they are able to identify an interesting system parameter under these conditions. It would be interesting to see if a hybrid of GEnBP and the ROAD-EnKF could be devised which would inherit strength from both. Such a method does not seem immediate however, and so we defer that to future work
>
> We have added a reference to this paper in the introduction, and restructured the introduction to position our paper with respect to it.
>
> > * Lin Z, Sun Y, Yin F, Thiéry A. "Ensemble Kalman Filtering-Aided Variational Inference for Gaussian Process State-Space Models." arXiv preprint arXiv:2312.05910, 2023.
>
> Thanks for this reference.
> Once again this exploits the EnKF in a state-space model, although this time to approximate the static transition operator in a Gaussian process state-space model.
> Despite a different setting to the Chen references, this once again targets a variational objective via an EnKf filtering approximation.
> This method does not look directly comparable for two reasons
>
> 1. it depends upon the system identification structure
> 2. This method does not target high-dimensional state spaces; as presented, it is unidimensional. Although expanding it to a low-dimensional vector space seems feasible, a high-dimensional kernel seems highly non-trivial; It would also be unlikely that this essentially nonparametric prior would be ideal for the systems we consider here, where there is a physics simulator available to constrain solutions, i.e. the $f$ mapping in this does not correspond closely to the $\mathcal{P}$ mapping in our work
>
> > * Girin L, Leglaive S, Bie X, Diard J, Hueber T, Alameda-Pineda X. "Dynamical variational autoencoders: A comprehensive review." arXiv preprint arXiv:2008.12595, 2020.
>
> We appreciate this reference for its comprehensive review of system identification methods. It underscores the breadth of the field, reinforcing the novelty of our work. Notably, this review and others like it highlight few attempts to address high-dimensional state spaces, leverage physical system simulators, or develop general message-passing algorithms for high-dimensional systems as our method does.

---

> > ### Comment · Reviewer_HdeM · 2024-11-26
> >
> > - Thank you for the clarification. I now have a clearer understanding of the paper's positioning and the difference between this paper and the DVAE series (including AD-EnKF & GPSSMs). I am satisfied with the author's response and improvements.
> >
> > - A small correction: In GPSSM, the latent state is not necessarily univariate. When extended to multiple dimensions, it is typically modeled with multiple independent GPs.
> >
> > - I will maintain the current score at this stage. Thank you.

---

> ### Author Response · Authors · 2024-11-21
> **Questions**
>
> ### Q1
>
> > Figure 2 is not cited in the text, which may confuse readers
>
> Thank you for pointing this fact out, which has been noted by other reviewers. We have updated the text to include a reference to that figure. In light of the fact that this figure has seemed to be a distraction to many readers, we propose to move it to the appendix, freeing up space in the body of the text for the many small edits that have been suggested.
>
> ### Q2
>
> > Consider put Figure 1 in the experimental section
>
> Thank you for this suggestion. We are sympathetic to to the reviewer's suggestion that the figure could be nearer to the experiment section.
> However, the readers who read the paper before submission have found the figure to be a helpful pedagogically as it allows an instant visualisation of the what the method does in fact do; as such, since it also plays a role in the introduction, we prefer to keep it there so that we may introduce the concepts pedagogically.
>
> ### Q3
>
> >  The paper highlights potential applications in fields like geospatial prediction and physical model inversion, which may involve real-time data. How does GEnBP handle streaming data, and what would be required to extend it to real-time applications with time-varying dependencies?
>
> Thank you for this question!
> Indeed one of the reasons that state filtering methods are popular in the geosciences is that they are able to handle streaming data.
> And one of the reasons that Belief-propagation methods are popular in SLAM is also that they are able to handle streaming data (see for example the work of Dellaert and Kaess).
> As such, GEnBP is also able to handle streaming data. We have not addressed this in the paper, as the paper already contains much material.
> In general, the SLAM literature on when to marginalise out a node is relevant here, as is the literature of optimal computation on belief propagation graphs. We are reluctant to add more material about advanced uses of belief propagation; rather out purpose in constructing a Belief-propagation extension to EnKF is precisely that all the handy features of belief propagation can be imported into the EnKF framework.
> This includes streaming data, but also causal reasoning, distributed computation, and so on.
> These are each interesting in their own right, but not necessary to build the basic operations.
>
> We have added a sentence to the introduction to add streaming data to the list of reasons that Belief Propagation algorithms are desirable.

---

> ### Author Response · Authors · 2024-11-27
> **Weaknesses part 3: Factor degree**
>
> Regarding the reviewer's question about the scaling with node degree: We wonder if the reviwer's opinion about the significance of this might be altered if we mention the fact, which is implicit but never actually stated in the paper, that the asymptotic complexity of GEnBP ~~never~~ rarely worse (asymptotically) than that of the classic GaBP? To see this, note that a DLR precision matrix may be converted to a dense covariance matrix at cost $\mathcal{O}(N^3 + N^2 D)$, and recovered at a cost of $\mathcal{O}(ND\log M+M^2(D+N))$ which is not a dominant term and converting back to the DLR form is of cost $\mathcal{O}(D^2\log N+N^2D+N^3)$ as per Appendix I. ~~These are both dominated by the $D^3$ terms in vanilla GaBP message calculations and thus do not contribute to the asymptotic complexity. As such, we need never pay an (asymptotic) complexity penalty for using GEnBP.~~
> Correction: if we do this every message passing iteration, then this does contribute the the complexity. A more accurate phrasing is that "the user does not need to pay a penalty for using GEnBP" --- they may trade off factor degree versus dimension to optimise cost.
>
> We have amended the text to account for this, modifying the end of section 3 to mention this, and referring the reader to Appendix I for details, where we expand upon this argument at length.

---

> ### Author Response · Authors · 2024-11-28
> **We have added non-system-identification experiments to the paper**
>
> We thank our reviewer for their engagement with the paper. May we draw your attention to Common Response 4, which explores a non-system-identification-type problem with an estimand of the scale of $10^6$ dimensions. We feel that this demonstrates applicability of the method to problems substantially outside the reach of off-the-shelf GPSSM and vaiational autoencoders, both in terms of problem structure and problem scale. The relevant part of the common response is included below for your convenience.
>
> >we contend that the broader significance lies in whether this method advances the frontiers of generic belief propagation problems that cannot be treated solely within the existing system identification literature.
> >
> >To address this, the new Appendix subsection (B.3) provides extensive explanations of the scope of a specific non-system-identification type. Specifically, we apply GEnBP to the problem of showcasing its applicability to generic belief propagation scenarios. The results demonstrate that GEnBP effectively handles a latent domain-adaptation problem for a high-dimensional neural network with high-dimensional outputs, thereby illustrating its broader utility beyond system identification.
> >
> >This addition is referenced in Section 4 (Experiments) as follows:
> >
> >  >  Although GEnBP is applicable beyond the system identification setting, this benchmark is chosen for its simplicity and popularity. More sophisticated graphical model problems are outlined in Appendix C, and in Appendix B.3 we demonstrate the utility of GEnBP applied to a different graphical structure: few-shot domain adaptation of a system dynamics emulator on unobserved states.
> >
> >In the new subsection, we demonstrate the applicability of GEnBP beyond system identification by applying it to a problem involving few-shot domain adaptation of a neural network emulator on unobserved states, which requires the unusual ability of the GEnBP method to scale to nodes with millions of dimensions: specifically, the millions of weights in a neural network. We set up a graphical model where a physical simulator predicts system states based on unknown parameters, and a Bayesian neural network (using the Fourier Neural Operator architecture) is trained to emulate this simulator using noisy observations without access to ground truth. By employing GEnBP to approximate the intractable posterior distribution of the neural network weights, we adapt the emulator to the new target problem. Our experiments on a 32×32 grid show that, after applying GEnBP to the first five of ten simulated timesteps, the adapted emulator closely follows the target distribution in subsequent timesteps, while the unadapted emulator diverges. This illustrates GEnBP's effectiveness in more complex, high-dimensional settings, highlighting its broader utility beyond traditional system identification tasks.[…]
> >
> >We are confident that this provides a solid response to the question about whether this paper is purely focused on system identification, and answers in the negative: this is a paper about belief propagation, and *the method is capable of solving interesting and non-trivial belief propagation tasks outside of system identification*. As such, this addition directly addresses the major outstanding issue raised by the reviewers, to the best of our ability to discern. We welcome commentary on this development.

---

### Official Review · Reviewer_dsRA · 2024-11-03

**Soundness:** 3
**Presentation:** 4
**Contribution:** 3
**Rating:** 6
**Confidence:** 3

**Summary:**

The authors introduce a novel method named Gaussian Ensemble Belief Propagation (GEnBP). They combine EnKF and GaBP together, in order to improve the efficiency in high-dimensional and non-linear cases. The method uses low-rank message-passing algorithms.

**Strengths:**

The authors try to use the strength of GaBP to solve (or I would like to say mitigate) the bad performance of EnKP in high-dim and non-linear problems. In the given examples, the new method works well. The computational complexity reduces largely for the dimensions of the data. It seems that the paper demonstrates superiority over existing methods in various benchmarks (but the figures are hard to read). I believe this method is potential for real-world applications for example in geospatial modeling.

**Weaknesses:**

1. Most important question for me: the scalability of the method to extremely large models, like weather simulations, remains to be fully demonstrated. Computational complexity still depends heavily on the degree of nodes in the graphical model, which means you must restrict the scale of the model. However, models in climate research always be extremely huge. How to choose N, how to choose K? If too small, can the model catches the nonlinearity well?
2. EnKF is somehow popular in climate or geophysical models, because of its computational complexity. But GaBP, in my impression, is a little bit outdated. For the nonlinear cases, we may use variational inference, or particle filter. Have you compared with them?
3. The innovation: GaBP mostly for low-dim problem. I am not sure it can help to improve the EnKF itself. More importantly, please see Bickson et al (2008) for the similar idea.
4. Even for small model, for example, Laplace seems better after consideration of trade-off. And Fig 2(c) seems not complete.

**Questions:**

Please see weaknesses.

---

> ### Author Response · Authors · 2024-11-21
> **Q2**
>
> > EnKF is somehow popular in climate or geophysical models, because of its computational complexity. But GaBP, in my impression, is a little bit outdated. For the nonlinear cases, we may use variational inference, or particle filter. Have you compared with them?
>
> This raises some interesting points. As our method is designed to generalize both GEnBP and GaBP, we of course have a vested interest in arguing that they are *both* outdated now that our work is available.
>
> Regarding your question about variational inference and particle filters, it is true that these methods represent significant advances in handling non-linear and complex models. We regard EnKF and GaBP as variational methods in a broad sense, as they both approximate solutions to complex inference problems, although they usually described differently. For further reading on this interpretation, we recommend:
>
> * Minka, T. P. (2005). [Divergence measures and message passing](https://tminka.github.io/papers/message-passing/). (brief)
> * Wainwright, M. J., & Jordan, M. I. (2008). [Graphical models, exponential families, and variational inference ](https://doi.org/10.1561/2200000001). (thorough)
>
> However, the essence of your query seems to b whether modern methods might render belief-propagation-type message passing obsolete. While it's true that methods like particle filters have been adapted for system identification and other applications, we know of few that address the goals of our method, to wit
>
> 1. message passing in graphical models with arbitrary structure, with
> 2. high-dimensional nodes
>
> Indeed, there are hundreds of distinct methods in the literature but few that that seem to offer promise to address this specific use case.
> Consider, for example, particle methods.
> It is true that particle filters can be extended to system identification, although not to more general graphs, e.g.
>
> Kantas, N., Doucet, A.,et al. (2015). [On Particle Methods for Parameter Estimation in State-Space Models.](https://doi.org/10.1214/14-STS511)
>
> However,particle methods  not generally be regarded as a strong candidate for high dimensional nodes, since particle methods are believed to typically scale poorly with dimensionality, see
>
> Snyder, C., Bengtsson, T., Bickel, P., & Anderson, J. (2008). Obstacles to High-Dimensional Particle Filtering. Monthly Weather Review, 136(12), 4629–4640. https://doi.org/10.1175/2008MWR2529.1
>
> The well-known difficulties in high-dimensional particle methods are a major reason for the popularity of EnKF methods in the geosciences, despite their limitations.
>
> As such, we have not pursued that specific comparison.
> Similar arguments can be made for other methods in the field.
>
> To address the task of comparing our belief-propagation-based method against non-belief-propagation-based methods, we have included comparisons with methods that are *not* based on graphical models, in particular, Laplace and Langevin Monte Carlo.
> These are also not state-filtering- or system-identification-specific, but generic approaches to Bayesian inference, and so we believe that they are indicatives of the trade-offs.
> Given that testing against all possible methods is prohibitive, would you identify any deficiencies in the choice of these baselines in particular?
>
> To enhance the clarity of our objectives and the relevance of message-passing methods, we propose the following modification to the manuscript, aimed at better emphasizing the unique advantages of belief propagation in complex inference scenarios:
>
> “Message-passing methods shine in various applications, including Bayesian hierarchical models (Wand, 2017), error-correcting codes
> (Forney, 2001; Kschischang et al., 2001), and Simultaneous Localization and Mapping (SLAM) tasks (Dellaert & Kaess, 2017; Ortiz et al., 2021), and inference distributed over many computational nodes Vehtari et al. (2019).”
>
> to
>
> “Message-passing methods shine various
> applications, including Bayesian hierarchical models (Wand, 2017), error-correcting codes (Forney,
> 2001; Kschischang et al., 2001), and Simultaneous Localization and Mapping (SLAM) tasks (Dellaert
> & Kaess, 2017; Ortiz et al., 2021). They possess various useful properties, such as permitting inference
> distributed over many computational nodes (Vehtari et al., 2019), domain adaptation (Bareinboim &
> Pearl, 2013) and natural means for estimaating treatment effects (Pearl et al., 2016).”
>
> We believe this adjustment will more accurately reflect the versatility and ongoing relevance of message-passing methods in the face of evolving computational challenges.
>
> We are grateful for the chance to discuss these aspects and look forward to further enriching our manuscript with these clarifications.

---

> ### Author Response · Authors · 2024-11-21
> **Q3**
>
> > The innovation: GaBP mostly for low-dim problem. I am not sure it can help to improve the EnKF itself. More importantly, please see Bickson et al (2008) for the similar idea.
>
> We thank the reviewer for mentioning Bickson's paper, which we agree is a useful work.
>
> The innovation of GEnBP lies in its hybrid approach: combining GaBP’s graph-aware message-passing with EnKF’s sample-based covariance representation. Unlike GaBP, GEnBP avoids explicit high-dimensional matrix operations by leveraging diagonal low-rank (DLR) representations (Section 3.1). The reference to Bickson (2008) is appreciated; we note that while both methods exploit Gaussian properties, Bickson’s work is focused on parallel computation for sparsely connected graphs, whereas GEnBP addresses broader inference problems in high-dimensional systems with non-linearities.
> There are of course connections; Bickson admits a high *overall* dimensionality, but the *local* (per-node) dimensionality is low. In our case, the local dimensionality is high, as well as the overall dimensionality.
> Nonetheless, while his work does not appear to scale to high dimensional node, we thank the reviewers for raising Bickson's work as a helpful demonstration of the usefulness of message-passing methods in scaling inference in that it shows that message-passing methods are powerful means of distributing computation. We note that Bickson's 2008 paper is included his 2009 thesis, which we cite as it has been useful in our work.
>
> Does that address your specific concerns regarding overlaps or novelty?
> If not, we would be happy to clarify our contributions more precisely,

---

> ### Author Response · Authors · 2024-11-21
> **Q4**
>
> ### Q4
>
> > Even for small model, for example, Laplace seems better after consideration of trade-off.
>
> We thank the reviewer for this observation about the experiment in section 4.1.
> Could they clarify which axes are in the trade-off in discussion here?
>
> As noted in our Common Response 1, we would like to take care to make clear what the intent of the comparison is.
> Firstly, the goal of section 4.1 and the associated figures 1 and 2 is to show the behaviour of the model on a simple problem whose solutions may be easily visualised.
> We do not expect the GEnBP to produce world-beating results on this problem, but rather to show that it works.
> Nonetheless, it provides a speed increase over the Laplace approximation, which is surprising for a method (message-passing inference) which is in many ways more general than the Laplace method.
>
> We note in passing that we included Figure 2 in the paper at the insistence of a previous reviewer, and we argue that since the current reviewers seem to agree with the authors, that its results are distracting, we propose to move this figure to the appendix, freeing up space in the body of the text for the many small edits that have been suggested.
>
> The more crucial test for hypothesis is the performance in the high dimensional physics example in section 4.2,  where the GEnBP method produces high quality results to produce results in domains where the posterior approximation used in the Laplace method would not fit in memory (Figure 3).
>
> To clarify this point, we propose to update the first sentence of section 4.1 to read:
>
> “The transport problem (Appendix A.2) is a simple, 1d nonlinear dynamical system chosen for ease of visualisation rather than high-dimensional performance.
> . States are subject to both transport and diffusion, where the transport term introduces
> nonlinearity. Observations are subsampled state vectors perturbed by additive Gaussian noise. Figure 1
> shows exemplary samples from the prior and posterior distributions induced by GaBP and GEnBP methods, and demonstrate a substantially improved ability for GEnBP to recover the posterior in this setting.
> Analysis of this contrived exampled is continued in Appendix A.2.1. where it is compared also against a global Laplace approximation.”
>
> >  And Fig 2(c) seems not complete.
>
> Thank you for the feedback, which other reviewers have noted as well.
> We have prepared a new version of the figure, and will post as new common response to all reviewer about this momentarily.
>
> We hope these responses address your concerns and clarify the contributions of our work. If additional clarification or experiments would strengthen the manuscript, we are eager to incorporate your suggestions.

---

> ### Author Response · Authors · 2024-11-21
> **Q1**
>
> > Most important question for me: the scalability of the method to extremely large models, like weather simulations, remains to be fully demonstrated. Computational complexity still depends heavily on the degree of nodes in the graphical model, which means you must restrict the scale of the model. However, models in climate research always be extremely huge. How to choose N, how to choose K? If too small, can the model catches the nonlinearity well?
>
> We thank the reviewer for their clarity in their priorities, and the opportunity to clarify our exact objective in this paper.
>
> We agree that scalability is a crucial consideration. The computational complexity of Gaussian Ensemble Belief Propagation (GEnBP) is analyzed in Section 3.4 and summarized in Table 2 of the manuscript. Notably, GEnBP avoids the \(O(D^3)\) scaling of GaBP for node-wise operations by leveraging ensemble-based approximations. Instead, its worst-case complexity scales as \(O(K^3N^3 + DN^2K^2)\), where \(K\) is the degree of a node, \(N\) is the ensemble size, and \(D\) is the dimension. In practice, \(N \ll D\), which substantially reduces computational burdens.
> Regarding the selection of \(N\), Figure 7 shows diminishing returns beyond \(N = 64\) for our benchmarks.
>
> The choice of \(K\) depends on the structure of the graphical model; we address this by considering sparsity patterns and using Forney factorization (Section 3.4) to manage node degree in dense graphs.
> Quantifying the set of all graph structures for which this method would dominate is an interesting question.
> Here we have refrained from claim that our method dominates in all scenarios, but that there exist scenarios where it is superior to existing methods; in particular the important case of the system identification problem used in the examples.
> We have amended the conclusion to reiterate this point, adding the text
>
> “Comparing overall computational cost of the algorithms is complicated and depends upon the graph structure.
> Our empirical results demonstrate the existence of problems where GEnBP is much more efficient than GaBP, but we have not characterised the full range of problems where this is the case.”
>
> Could you identify specific scenarios or data where scalability is a concern? For example, are there specific \(K\) or \(D\) ranges in your envisioned climate applications?

---

### Author Response · Authors · 2024-11-20
**Common response 1**

Thank you for your detailed reviews and suggestions for comparing our Gaussian Ensemble Belief Propagation (GEnBP) method with various existing techniques. We value your insights, which help clarify our method's place within the broader research landscape.

There are  themes which have arisen generally in the reviews which we feel deserve a general response, which we would classify as “general questions about the scope of the claims.”

Several suggested comparisons, including autodifferentiable EnKF and dynamical variational autoencoders, are indeed valuable within system identification and state filtering. However, GEnBP targets a broader application set within the "belief propagation" methods, applicable to more general graphical models, as exemplified in Figure 8 of the appendix.

We acknowledge that our initial problem *is* a problem of system identification type, which we have selected because of its popularity. It would be unlikely for GEnBP to dominate in general in system identification over all possible methods designed specifically for that task.
For this case, demonstration that the GEnBP method, devised for arbitrary graphs, happens to be competitive with existing methods customised for system identification problems is, we argue, sufficient to establish our claim as made.

Unlocking efficient inference in high-dimensional graphical models is a high value goal for all the reasons that belief propagation is a valuable tool in the first place; e.g. we can can estimate intervention effects, impute missing values, distribute computation over many nodes and so on.

That said, we are sympathetic to the idea that if we wish to make a claim about a more general graphical model structure, there is a case for including example problems more general than the system identification problem. We have made the tactical choice not to prioritise that because the methodology section of the paper is already dense, and the appendices are already twice as long as the paper itself. We are happy to consider this for future work.

To align our manuscript with these clarifications, we propose the following revisions:

1. **Introduction and Discussion:** We will clarify GEnBP’s role within belief propagation, differentiating it from methods geared specifically towards system identification or state filtering.

2. **Scope Clarification:** We will explicitly note that GEnBP is not designed to universally surpass all other methods on all graphical structure but to complement the existing toolkit for handling high-dimensional inference problems.

3. **Emphasizing Novelty:** The integration of ensemble methods with belief propagation in complex models is a significant innovation of our work, and we will highlight this contribution more prominently.

We appreciate the opportunity to refine our work and believe these revisions will make the paper's objectives clearer.

---

### Author Response · Authors · 2024-11-22
**Common response 2: Manuscript updated.**

We are grateful to the reviewers for their thorough evaluations and insightful comments on our manuscript.
We have prepared a revision. Changes are also colorized in the PDF. This pushes it over the 10 page limit, but when the change markers are removed the paper is still 10 pages long.

### 1. Scalability and Computational Complexity *(dsRA, HdeM)*

- **Clarified Scalability Considerations**: In response to **dsRA** and **HdeM**'s concerns about the scalability of our method to extremely large models and highly connected graphs, we have:

  - **Expanded the discussion in Section 3.4** to detail the computational complexity of GEnBP, emphasizing that while it avoids the \(O(D^3)\) scaling of GaBP, its complexity depends on the graph structure and node degree.

  - **Added to the conclusion**: “Comparing overall computational cost of the algorithms is complicated and depends upon the graph structure. Our empirical results demonstrate the existence of problems where GEnBP is much more efficient than GaBP, but we have not characterized the full range of problems where this is the case.”

### 2. Comparison with Related Methods *(dsRA, HdeM, Mt1V)*

- **Expanded Literature Review**: To address the absence of comparative analysis noted by **dsRA**, **HdeM**, and **Mt1V**, we have:

  - **Added references** to related works, including:  Chen et al.'s autodifferentiable EnKF methods, Lin et al.'s Ensemble Kalman Filtering-Aided Variational Inference, Girin et al.'s review on dynamical variational autoencoders, Methods like Integrated Nested Laplace Approximation (INLA) and particle Monte Carlo methods.

  - **Discussed the positioning of our work** relative to these methods, highlighting the unique aspects of GEnBP in handling high-dimensional nodes within arbitrary graphical model structures.

  - **Clarified why system identification methods were not directly compared**, explaining limitations in high-dimensional settings and differences in application focus.

### 3. Clarification of Novelty and Contribution *(dsRA)*

-  In response to **dsRA**'s mention of Bickson et al. (2008), we have **Clarified the novelty of GEnBP**, explaining that while Bickson's work focuses on parallel computation in sparsely connected graphs with low-dimensional nodes, our method addresses high-dimensional systems with non-linearities and high-dimensional nodes, and in fact already cites Bickson.

### 4. Improved Figures and References* *(dsRA, QpY9, Mt1V)*

- **Updated Figure 2**: Responding to **dsRA**, **QpY9**, and **Mt1V**, we have prepared a new version of Figure 2, addressing incomplete data and enhancing readability.

- **Moved Figure 2 to the appendix** and updated reference to streamline the main text, as it seemed to distract readers from the core content.

- **Revised Section 4.1**: We clarified that the purpose of the experiments was to demonstrate the functionality of GEnBP on simple, visualizable problems rather than claiming superiority in all scenarios.

 - **Explained missing log-likelihood values**: We clarified that missing values in Figure 2 are due to non-positive-definite covariance estimates, making the log-likelihood undefined. Our revised plotting method exludes undefined values and so has fewer missing ranges on the plot.

- **Corrected results in Figure 3**: For **QpY9**, we re-examined the high-viscosity results and found that the problem becomes ill-posed in this regime. We have updated the figure and discussion accordingly.

### 5. Explanation of Methodological Details *(Mt1V)*

- we have  **Explained the role of particle representation** in GEnBP, highlighting that it serves a similar function to the linearization step in traditional GaBP and is essential for handling non-linearities in high-dimensional nodes with reference to App. I and  Figure 11

 - **Acknowledged the potential benefits** of techniques like covariance localization and inflation, commonly used in EnKF, for enhancing GEnBP's performance in the conclusion


### 7. Stylistic and Language Improvements *(QpY9)*

- **Revised the Manuscript for Clarity**:

  - **Corrected grammatical errors and improved punctuation**, addressing **QpY9**'s concerns.

  - **Ensured consistency in terminology and notation** throughout the paper.

### 8. Future Work and Limitations *(All Reviewers)*

- **Acknowledged Method Limitations**:

  - **Emphasized that GEnBP is not universally superior**, and its performance depends on the specific problem structure and graph topology.

- **Outlined Directions for Future Research**:

  - **Suggested integrating techniques** from both GaBP and EnKF literature, such as graph pruning, robust distributions, and the aforementioned covariance localization and inflation.

  - **Proposed exploring hybrids** of GEnBP with other methods like INLA or particle methods, as suggested by **Mt1V**.

---

### Author Response · Authors · 2024-11-26
**Common response 3: new revision.**

We have uploaded a new revision of the manuscript. Differences from the initial version are colorized. This one includes fixes for typos introduced in previous revisions, and an expanded explanation on the rationale for performing the specific task of system identification with a generic belief propagation method.

To paraphrase our own conclusion, we have shown that traditional belief propagation methods are leaving performance "on the table" by not exploiting the DLR structure and convenient conditioning produced by ensemble methods, and moreover, we suspect that this innovation is largely "pluggable" into the existing ecosystem of practices in this domain. On that basis, we believe that we are able to provide immediate quality-of-life improvements to researchers applying machine learning to spatiotemporal data, and we hope that this work is published to disseminate that improvement.

If we are blessed with luck, some bonus experiments may land in the paper before the deadline. However, we believe that the paper capable of standing on its own merits as-is. We welcome any final engagement with or requests from our reviewers on what might induce them to share this opinion.

---

### Author Response · Authors · 2024-11-27
**Common response 4: New experiment on domain adapatation**

We would like to thank the reviewers for their thoughtful comments and for highlighting areas where our manuscript could be improved. As noted before, we believe that our reviewers have, to varying degrees, assessed the merit of this paper based on whether it represents the state-of-the-art (SOTA) in the system identification problem. While we acknowledge that this is an interesting application, we contend that the broader significance lies in whether this method advances the frontiers of generic belief propagation problems that cannot be treated solely within the existing system identification literature.

To address this, the new Appendix subsection (B.3) provides extensive explanations of the scope of a specific non-system-identification type. Specifically, we apply GEnBP to the problem of showcasing its applicability to generic belief propagation scenarios. The results demonstrate that GEnBP effectively handles a latent domain-adaptation problem for a high-dimensional neural network with high-dimensional outputs, thereby illustrating its broader utility beyond system identification.

This addition is referenced in Section 4 (Experiments) as follows:

>Although GEnBP is applicable beyond the system identification setting, this benchmark is chosen for its simplicity and popularity. More sophisticated graphical model problems are outlined in Appendix C, and in Appendix B.3 we demonstrate the utility of GEnBP applied to a different graphical structure: few-shot domain adaptation of a system dynamics emulator on unobserved states.

In the new subsection, we demonstrate the applicability of GEnBP beyond system identification by applying it to a problem involving few-shot domain adaptation of a neural network emulator on unobserved states, which requires the unusual ability of the GEnBP method to scale to nodes with millions of dimensions: specifically, the millions of weights in a neural network. We set up a graphical model where a physical simulator predicts system states based on unknown parameters, and a Bayesian neural network (using the Fourier Neural Operator architecture) is trained to emulate this simulator using noisy observations without access to ground truth. By employing GEnBP to approximate the intractable posterior distribution of the neural network weights, we adapt the emulator to the new target problem. Our experiments on a 32×32 grid show that, after applying GEnBP to the first five of ten simulated timesteps, the adapted emulator closely follows the target distribution in subsequent timesteps, while the unadapted emulator diverges. This illustrates GEnBP's effectiveness in more complex, high-dimensional settings, highlighting its broader utility beyond traditional system identification tasks.

The content of Section B.2 is approximately two pages and adds two additional diagrams (Figures 8 and 10); we refer the reviewers to the manuscript for details. The figures are draft quality and will require minor adjustments for the camera-ready copy, but the data should be unambiguous.

We are confident that this provides a solid response to the question about whether this paper is purely focused on system identification, and answers in the negative: **this is a paper about belief propagation, and the method is capable of solving interesting and non-trivial belief propagation tasks outside of system identification**. As such, this addition directly addresses the major outstanding issue raised by the reviewers, to the best of our ability to discern. We welcome commentary on this development.

We appreciate the reviewers' insights and believe that these enhancements strengthen our manuscript by highlighting the versatility and broader impact of GEnBP in the field of belief propagation.

---

### Meta-Review · Area_Chair_Fjwf · 2024-12-20

**Metareview:**

The authors introduce a novel method named Gaussian Ensemble Belief Propagation (GEnBP). They combine EnKF and GaBP together, in order to improve the efficiency in high-dimensional and non-linear cases. The method uses low-rank message-passing algorithms.

The paper received four reviews, and all reviewers agreed that the paper was above threshold for acceptance, with two arguing for acceptance. The authors satisfied several reviewers'' concerns and comments, with some raising their scores as a result. Working in some of that new information, space permitting, would make for a stronger final version.

**Additional Comments On Reviewer Discussion:**

Three of the reviewers were satisfied with the responses during the discussion period and two chose to raise their scores. We remind the authors that it's common for a reviewer not to respond to the rebuttal, so releasing them in frustration from this non-existent requirement was an unnecessary flourish.

---

### Decision · Program_Chairs · 2025-01-22

Accept (Poster)